# Participatory modelling for poverty alleviation using fuzzy cognitive maps and OWA learning aggregation

**Konstantinos Papageorgiou**[1]\*, **Pramod K. Singh**[2], **Elpiniki I. Papageorgiou**[3,4],
**Harpalsinh Chudasama**[2], **Dionysios Bochtis**[4], **George Stamoulis**[5]

**1** Department of Computer Science and Telecommunications, University of Thessaly, Lamia, Greece,
**2** Institute of Rural Management Anand (IRMA), Gujarat, India, **3** Department of Energy Systems, Faculty of
Technology, University of Thessaly, Larissa, Greece, **4** Institute for Bio-economy and Agri-technology (iBO),
Center for Research and Technology-Hellas (CERTH), Thermi, Thessaloniki, Greece, **5** Department of
Electrical and Computer Engineering, University of Thessaly, Volos, Greece

\* konpapageorgiou@uth.gr

**Data Availability Statement:** All relevant data are within the paper and its Supporting Information files.

## Abstract

Participatory modelling is an emerging approach in the decision-making process through which stakeholders contribute to the representation of the perceived causal linkages of a complex system. The use of fuzzy cognitive maps (FCMs) for participatory modelling helps policy-makers develop dynamic quantitative models for strategising development interventions. The aggregation of knowledge from multiple stakeholders provides consolidated and more reliable results. Average aggregation is the most common aggregation method used in FCMs-based modelling for weighted interconnections between concepts. This paper proposes a new aggregation method using learning OWA (ordered weighted averaging) operators for aggregating FCM weights assigned by various stakeholders. Besides, we report a comparative analysis of 'OWA learning aggregation' with the conventional average aggregation method, while evaluating the theory of change for the world's most extensive poverty alleviation programme in India. The results of the FCMWizard web-based tool show that the proposed method provides an opportunity to policy-makers for evaluating outcomes of proposed policies while addressing social resilience and economic mobility.

## 1. Introduction

Poverty is not merely a low income but multidimensional phenomenon involving lack of income or consumption, food insecurity, high vulnerability to risks, low human capital, unequal social relations, and powerlessness [1]. Hence, poverty eradication involves complex interactions within socio-economic systems. Understanding such complex interactions requires a participatory modelling approach that can be implemented by fuzzy cognitive maps.

In participatory modelling, stakeholders allow decision-makers to understand meaningful interactions occurring inside a complex system and contribute valuable first-hand knowledge for supporting decision-making, policy formulation, regulation, and management purposes

**Funding:** The author(s) received no specific funding for this work.

**Competing interests:** The authors have declared that no competing interests exist.

[2]. Fuzzy cognitive maps (FCMs) introduced by Kosko [3], are fuzzy directed graphs that can model any real-world system. FCMs can easily integrate diverse human knowledge and adapt to a particular domain [4]. FCM is a common participatory modelling methodology due to causal linkages of the system [5]. In recent years, FCMs have been applied extensively in multiple domains due to their simple model structure and ease of use [6, 7]. It is no wonder that they have become a powerful soft computing modelling tool [8]. Moreover, due to their promising modelling capabilities, a significant number of FCM extensions has been applied for modelling of causal relationships between concepts, as reported in the extant literature [9–11]. Among them, Neuro-Fuzzy Cognitive Map (NFCM) was emerged as a recent fuzzy modelling approach that uses empirical data in the form of fuzzy rules to determine the weights of relationships among concepts [12]. The training of the developed network is accomplished when the desired values of its parameters are obtained. In all cases, the FCMs-based participatory modelling approach attempts to capture the causal relationships within complex systems by using the views and perceptions of stakeholders. Doing so can reduce conflicts among stakeholders by capturing different inter-sectorial synergies and tradeoffs, helping them to reach consensus in a complex policymaking environment.

One may aggregate FCMs constructed by experts, stakeholders or both, to produce a combined FCM model, enriched with the knowledge of the experts and stakeholders. Typically, the input models to be aggregated are developed by multiple domain experts [4], to improve the reliability of the final FCM model. That is, the FCM model becomes less susceptible to potentially erroneous beliefs of a particular expert, helping to bridge knowledge discrepancies among the participants. There are mainly two techniques for combining multiple FCMs into a single collective FCM model, commonly used in a wide range of real-life problems [13]. These techniques are the weighted average method and the OWA (ordered weighted averaging) aggregation method introduced by Yager [14].

In the respective literature, the weighted average method has been used as a benchmark method for FCM aggregation purposes and is particularly useful for descriptive causal relationships, linguistically expressed through fuzzy logic [15]. The collective FCM model is constructed by averaging numerical values for a given interconnection [16], whose rules are aggregated with the help of fuzzy operators, while the overall output is elaborated using the weighted average of the output of each rule [8].

On the other hand, the application of OWA operators for the aggregation of individual FCMs can be found in various research papers in the literature. Lv and Zhou [17] introduced the OWA operators in an FCM framework also highlighting the problems related to the determination of weights for the OWA-based aggregation. Also, a distance-based OWA operator was used in [18] to rank the scenarios that depend on the risk preferences of decision-makers. Recently, a study on the exploration of an alternative FCM aggregation method using OWA operators was conducted by Papageorgiou et al. [19, 20].

The current work extends the previous study by introducing the weights from learning OWA operators for the aggregation of FCM links, as noticed that there is no such work previously conducted in FCM-based participatory modelling. The proposed OWA-based aggregation approach that involves learning operators to define weights between the concepts, complies with broad applicability and is highly effective in the case of aggregated views of a large number of participants. The applicability of the proposed methodology is demonstrated through a complex real-life development intervention with a large number of participants being involved in designing the FCM model.

A Java-based tool called OWA-FCM, was used for implementing the OWA operators. This flexible and user-friendly tool was developed to provide the research community with an automated method to aggregate a large number of individual FCMs. Besides, a new software tool,

namely FCMWizard (www.fcmwizard.com), has been recently presented to provide FCM researchers and policymakers with a simulation function, necessary to make decisions and perform policy simulations [21]. This web-based tool with simulation and learning capabilities, allows the construction of FCMs by using either the knowledge of experts or the provided data and is available under request.

This study is dedicated to the following: (i) to apply the quasi-qualitative method of fuzzy cognitive mapping to model communities' perception, (ii) to aggregate a large number of individual FCMs with a new OWA-based aggregation approach, (iii) to integrate the level of confidence of stakeholders in the aggregation process, and (iv) to study various scenarios using FCM-based simulation process implemented by the new FCMWizard tool. National and state-level implementers and representatives of community-based organisations (CBOs) of the DAY-NRLM (*Deendayal Antyodaya Yojana*-National Rural Livelihoods Mission), the world's most extensive poverty alleviation programme, participated in this study.

## 1.1. Theory of change for poverty alleviation

The DAY-NRLM is a community institutions-based poverty alleviation programme sponsored by the Government of India. The predominant focus of the DAY-NRLM is to reduce socio-economic poverty and improve the quality of life of the vulnerable and poor rural communities. The Jammu and Kashmir State Rural Livelihood Mission (JKSRLM) implemented the programme in the State under the name '*Umeed*', which means 'hope'.

Influential community-based organisations (CBOs), capacity building exrcises involving CBOs and their members, access provision to financial resources to CBOs and their members, and livelihood diversification and enterprise development form the pillars of the programme, as illustrated in Fig 1. The successful execution of these building blocks of the programme is likely to make the CBOs effective in programme delivery. Building strong CBOs for universal social mobilisation and the improvement of the governance of CBOs is expected to ensure access to financial resources in terms of a revolving fund, a community investment fund, and an SHG-bank linkage. The capacity building of community members and CBOs is conducted through community service providers. Diversification of livelihoods of the communities and enterprise development through skill development, value addition, and value chain infrastructure are other vital pieces of the programme. The institutional effectiveness of the CBOs successfully delivers its different set of interventions.

The principal intermediate outcomes of the programme include socio-economic inclusion, better governance, greater access to formal financial institutions, information, technology, market, and entitlements, among other things. The final outcomes of the programme include greater social harmony, asset creation, higher-income, better health, hygiene & sanitation, and better awareness & access to education, and less vulnerability, among other things, in conjuction with greater social, economic, and political empowerment of women. These outcomes are interconnected and provide feedback to each other. Ultimately, these outcomes will lead to the final delivery of the programme, i.e. decrease in socio-economic poverty and improvement in the quality of life. The theory of change helps us to develop input vectors for policy scenarios.

The innovations and contributions of this paper have been summarised as follows:

- Presenting a new aggregation method for FCMs based on OWA learning operators.

- Integrating the confidence level of the stakeholders in the aggregation process.

- Performing scenario analysis using a newly developed FCMwizard tool.

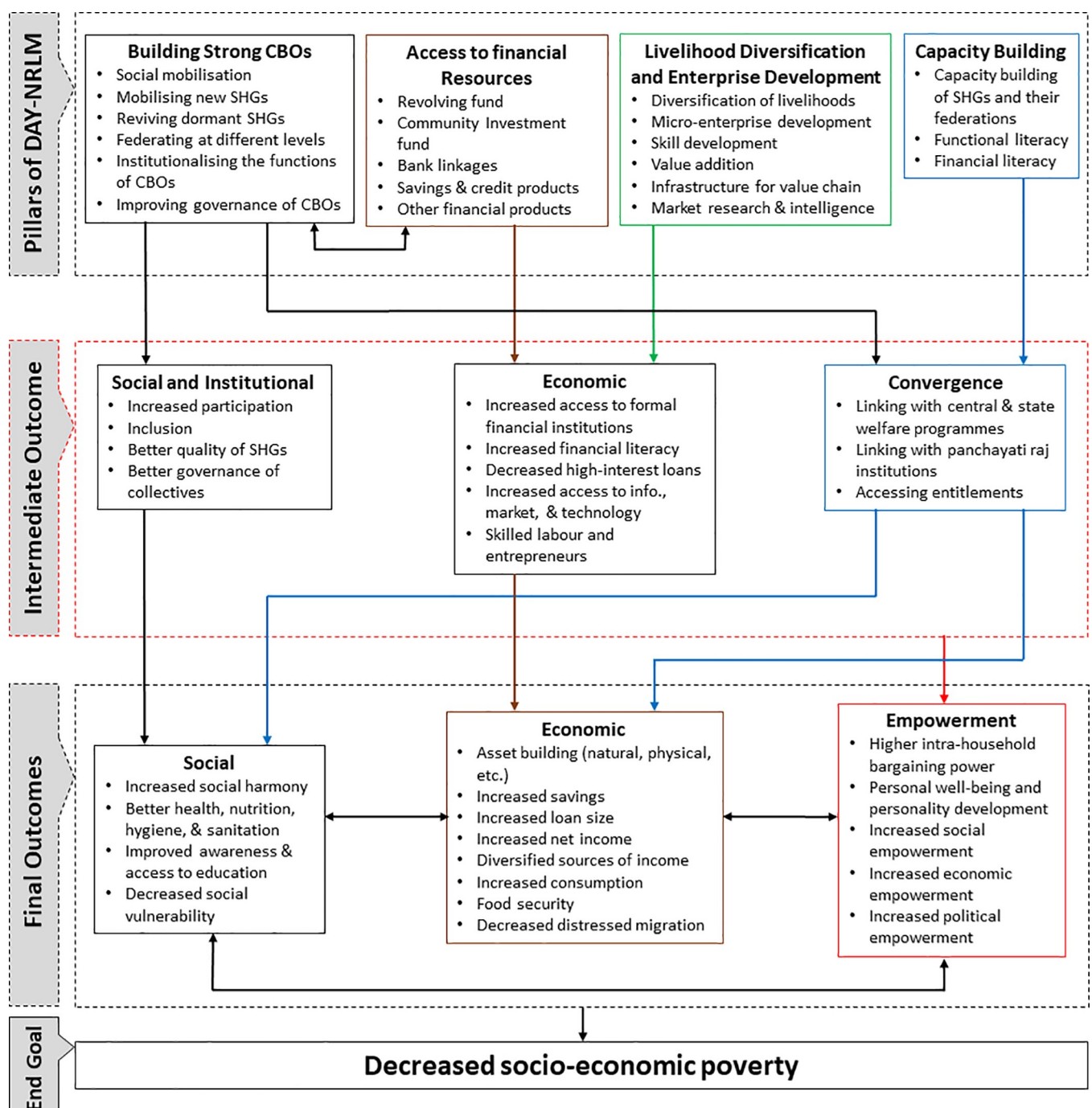

**Fig 1. Theory of change for alleviation of socio-economic poverty in Jammu and Kashmir.** This figure was developed in consultation with the programme implementers.

- Comparing the new OWA-based FCM aggregation method with the benchmark weighted average method.

- Demonstrating the use of aggregated OWA-FCM in the generation of policy scenarios.

The structure of this paper is as follows: Section 2 describes the methodology used to develop FCMs, the main aspects of OWA operators along with the proposed algorithm based on the weights of OWA operators for aggregating FCM weights. Section 3, includes the FCM-based scenario analysis and the conducted simulations. In the following section (section 4), all the results and discussions derived from the application of the OWA aggregation method, scenario analysis, and the sensitivity analysis are elaborated. In section 5, authors discuss the key remarks drawn, regarding the performance of the two aggregation methods. In the end, section 6 summarises the conclusions of the study.

## 2. Methodology for designing fuzzy cognitive maps

Fuzzy cognitive maps introduced by Kosko [3, 22] as fuzzy-graph structures to represent causal reasoning use a soft computing technique, which combines fuzzy logic and neural networks [23]. They can meet the needs of intelligent systems for the representation of knowledge and reasoning with an ease [24, 25].

An FCM encases several nodes (i.e. concepts) representing different elements of a particular system and directed connections (i.e. links) that establish causal relationships between the concepts [26]. A concept in an FCM is indicated by $C_i$, $i = 1,–...,N$ (N represents the total number of concepts in an FCM). The links have been labelled with values (i.e. weights) within the range [0,1] or [-1,+1], which shows the strength of the impact between the concepts [9]. Weights of the connections between concepts form a weight value matrix $E\ NxN$ called an adjacency matrix, where each element of the matrix $w_{ij}$ takes values in the range [-1,+1] [27]. Fig 2 illustrates a fuzzy cognitive map and its corresponding weight matrix for a randomised case study.

### 2.1. Obtaining cognitive maps from the participants

We adopted a mixed-concept design approach for obtaining cognitive maps from the participants, which involves an open-concept design followed by a closed-concept design.

**2.1.1. Expert-based FCM using an open-concept design.** We involved 31 national and state-level implementers in a group discussion regarding the impacts of DAY-NRLM interventions. It prompted them to identify factors responsible for achieving the outcomes of the programme. They identified 20 categories of main concepts and 129 sub-concepts that were then displayed over a whiteboard. After that, we demonstrated to the participants how to draw a fuzzy cognitive map citing impacts of deforestation, a neutral problem domain, as an example. Once the participants understood the process of constructing a cognitive map, we asked them to draw fuzzy cognitive maps by providing causal links to each main concept. Fig 3 represents the main concepts and their causal links identified by the experts. We requested the participants to assign weights of relationships between the concepts on a scale of 1–10 for each link. Ten (10) denoted the highest strength and one (1) the lowest [28]. We also asked them to assign level of confidence of their weights on a scale of 1–5 (1 = Very low confidence; 2 = Low confidence; 3 = Medium confidence; 4 = High confidence; and 5 = Very high confidence) for each link.

**2.1.2. Preparation of study protocol based on the expert-based model.** We prepared a protocol (S1 Fig) depicting all the 20 categories of main concepts and 129 sub-concepts generated by the experts (national and state-level programme implementers). We also showed the links identified by the experts in the protocol. The Research Ethics Committee of the Institute of Rural Management Anand (IRMA) approved this protocol. This Expert-based FCM model shown below (Fig 3) will be compared with the FCM models, derived from the aggregation processes, to help this study meet its objectives.

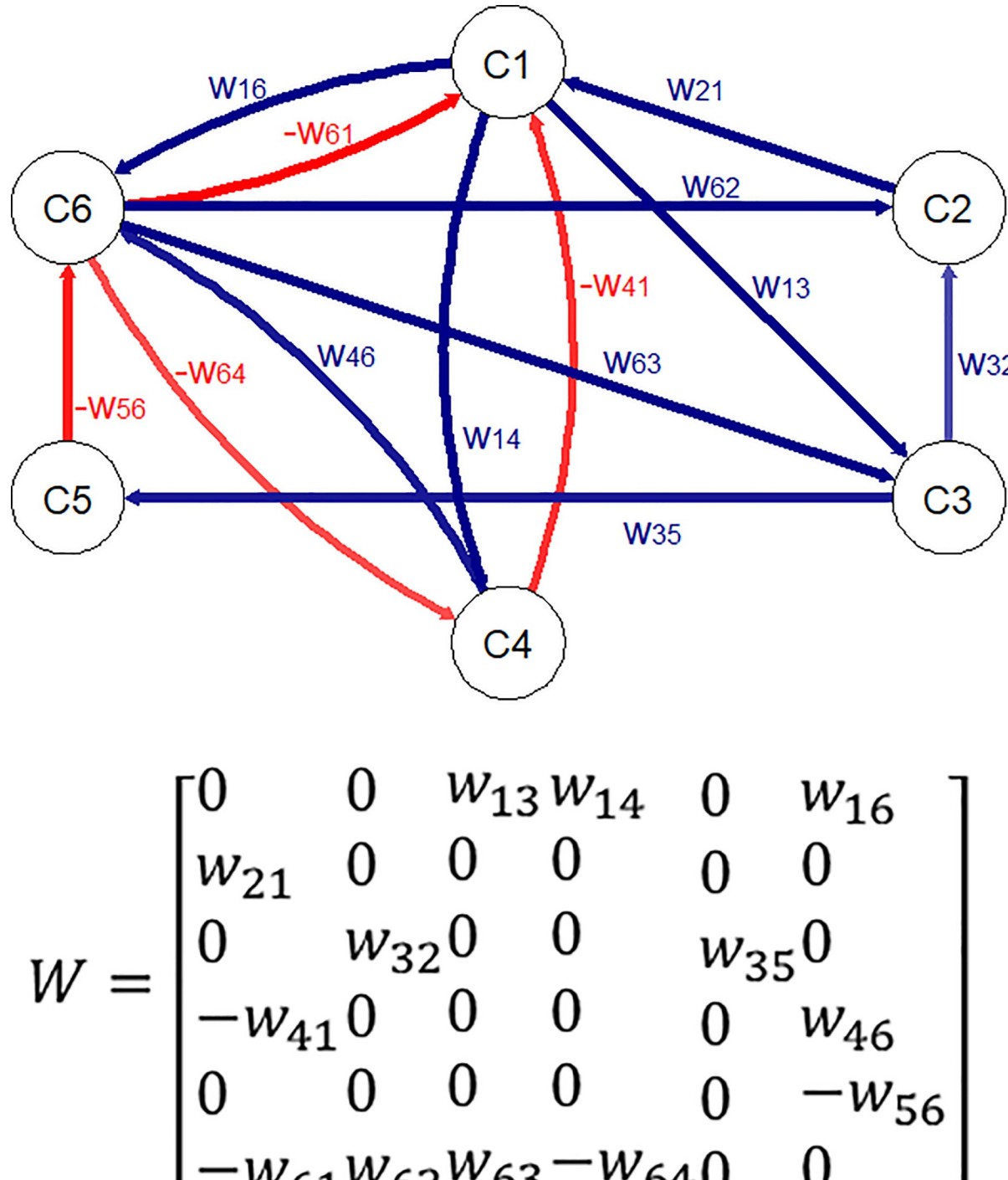

**Fig 2.** Fuzzy cognitive map (left) and the correspondent adjacency weight matrix (right).

**2.1.3. Closed-concept design from the community-level stakeholders.** We went on to administer the protocol to four different groups of stakeholders from DAY-NRLM. We asked them to construct FCMs, i.e., functionaries of Self-Help Groups (SHGs), their federations [Village Organisations (VOs) and Cluster Level Federations (CLFs)], and Community Resource

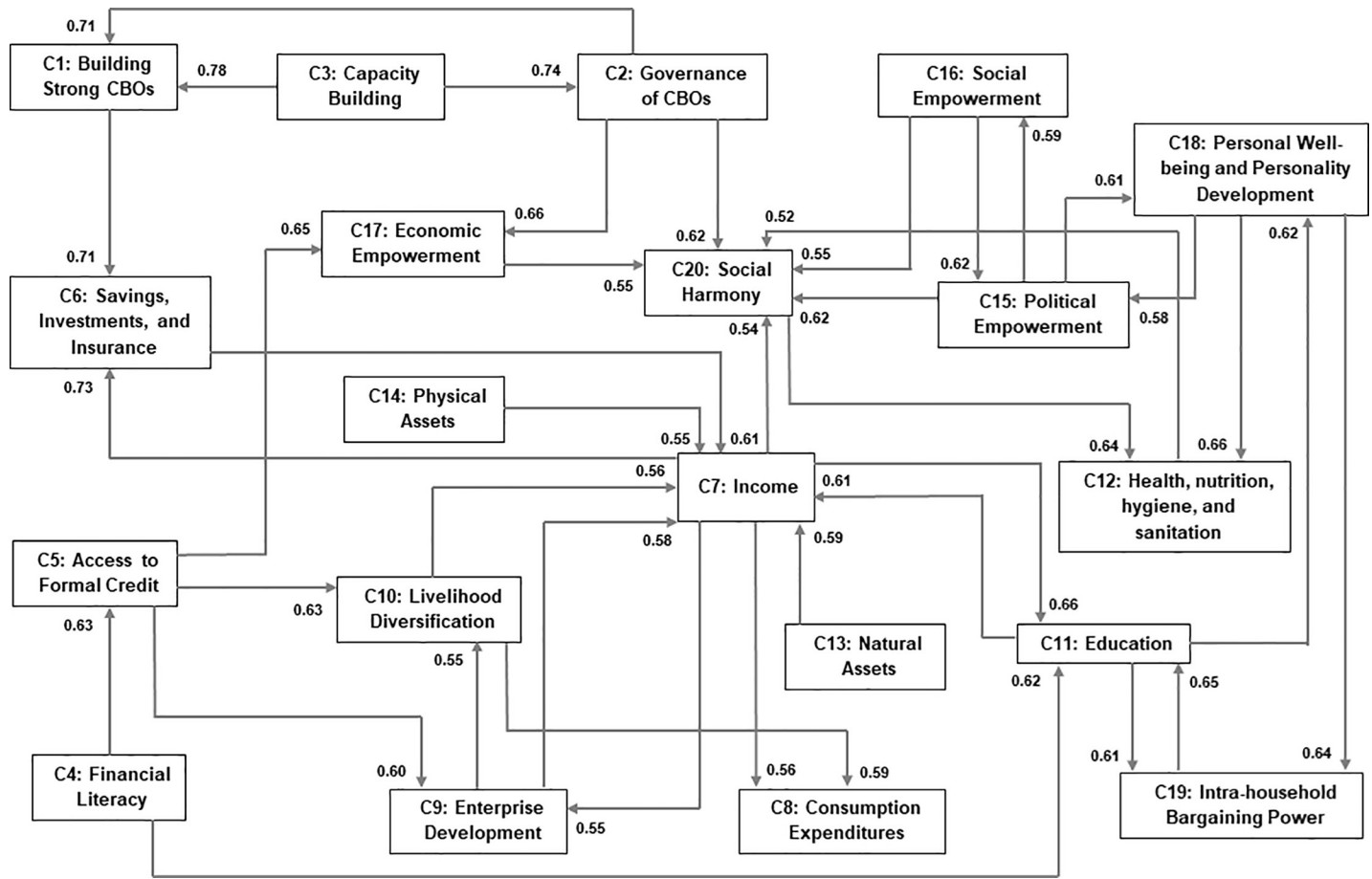

**Fig 3. Expert-based FCM model.**

Persons (CRPs). All the community stakeholders were women. A total of 179 fuzzy cognitive maps were obtained from over 600 participants during the FCM exercise. These participants were selected randomly from the programme participants. SHG and VO functionaries were divided into smaller sub-groups, with each sub-group comprising four to five members, to construct FCMs. On the other hand, the CLF functionaries and CRPs drew the FCMs individually.

During this exercise, every community group/ individual stakeholder provided weights to the individual links on a scale of 1–10 and the level of confidence for such weights on a scale of 1–5 as discussed earlier. The participants were also asked to draw new links between the categories, depending on what they believed. However, none of them created any additional link. Each group made a presentation to the researchers after having concluded construction of the FCM.

## 2.2. Coding individual cognitive maps and confidence level values into adjacency matrices

We coded each fuzzy cognitive map into separate Excel sheets with concepts listed in vertical $C_i$ and horizontal axes $C_j$; this formed a square adjacency matrix [28–32]. The positive wording of all the concepts warranted only positive values for the relationships. Accordingly, the weight values were coded into the adjacency matrix only when there was a connection between two

given concepts [28]. The weights given to each link were normalised between 0 and +1 (if the values +5, then they are normalised +0.5) while coding into the adjacency matrix [30–32]. Similarly, the confidence level values were also normalised between 0 and 1 (for example, value 1 corresponds to 0.2, value 2 to 0.4, and value 5 corresponds to 1).

## 2.3. Producing individual links (L) and confidences and links (C) matrices

Each adjacency matrix consists of the normalised weights of the individual cognitive map named as adjacency links (L) matrix, and the corresponding individual FCM model called as FCM (L) model. Taking one step further from the adjacency matrices related to each group or individual participants, we multiplied the normalised weight value of every existing link with its normalised confidence level value producing the adjacency confidence and links (C) matrix that corresponds to the individual FCM (C) model.

We created two FCM models based on each map: one for links (L) and another for confidences and links (C). For the second FCM model, we produced the final value (strength) of every connection by multiplying the weight of every connection with the value of the confidence level assigned by the participants to the corresponding links. We ended up building a new FCM model that seemed to represent participants' opinions with high reliability.

## 2.4. Aggregating individual cognitive maps from each group producing aggregated links (L) and confidences and links (C) FCMs

In this step, all individually coded cognitive maps were aggregated and additively superimposed using the two aggregation methods (the Average and the OWA) [20]. An overall group FCM (Collective-FCM) was produced from each of these methods, separately for both links (L) and confidences and links (C). The obtained Collective-FCM represents the perception of all the stakeholders (SHG, VO, CLF, and CRP) and includes all the concepts from all individual cognitive maps. Thus, two collective-FCMs were created for every group of participants for each aggregation method; the Average-FCM (L) and OWA-FCM (L), referring to the FCMs that include the weight value for every single link, and the Average-FCM (C) and OWA-FCM (C) where every link value between two nodes was the result of the multiplication of its weight value with its corresponding confidence level value. Fig 4 depicts a visual representation of the aggregation process.

**2.4.1. Average aggregation.** Kosko [3] suggested the average aggregation method for aggregating a large number of FCMs consisting of the same or different concepts (representing different variables, status, parameters, among other things.). Considering that $n$ experts assign a weight value $w_{ij}$, between the nodes $C_i$ and $C_j$ on individual FCMs with the same number of concepts, then the aggregated weight $w_{ij}^{(ave)}$ between these nodes may be defined as the average value of the $n$ weights $w_{ij}$:

$$w_{ij}^{(ave)} = \frac{w_{ij}^{(1)} + w_{ij}^{(2)} + \cdots + w_{ij}^{(n)}}{n} \tag{1}$$

**2.4.2. OWA aggregation.** An OWA operator of dimension $n$ is a mapping:

$$f : R^n \rightarrow R$$

that includes a correlated vector of weights $W$

$$W = [w_1 \ w_2 \ldots w_n]^T \tag{2}$$

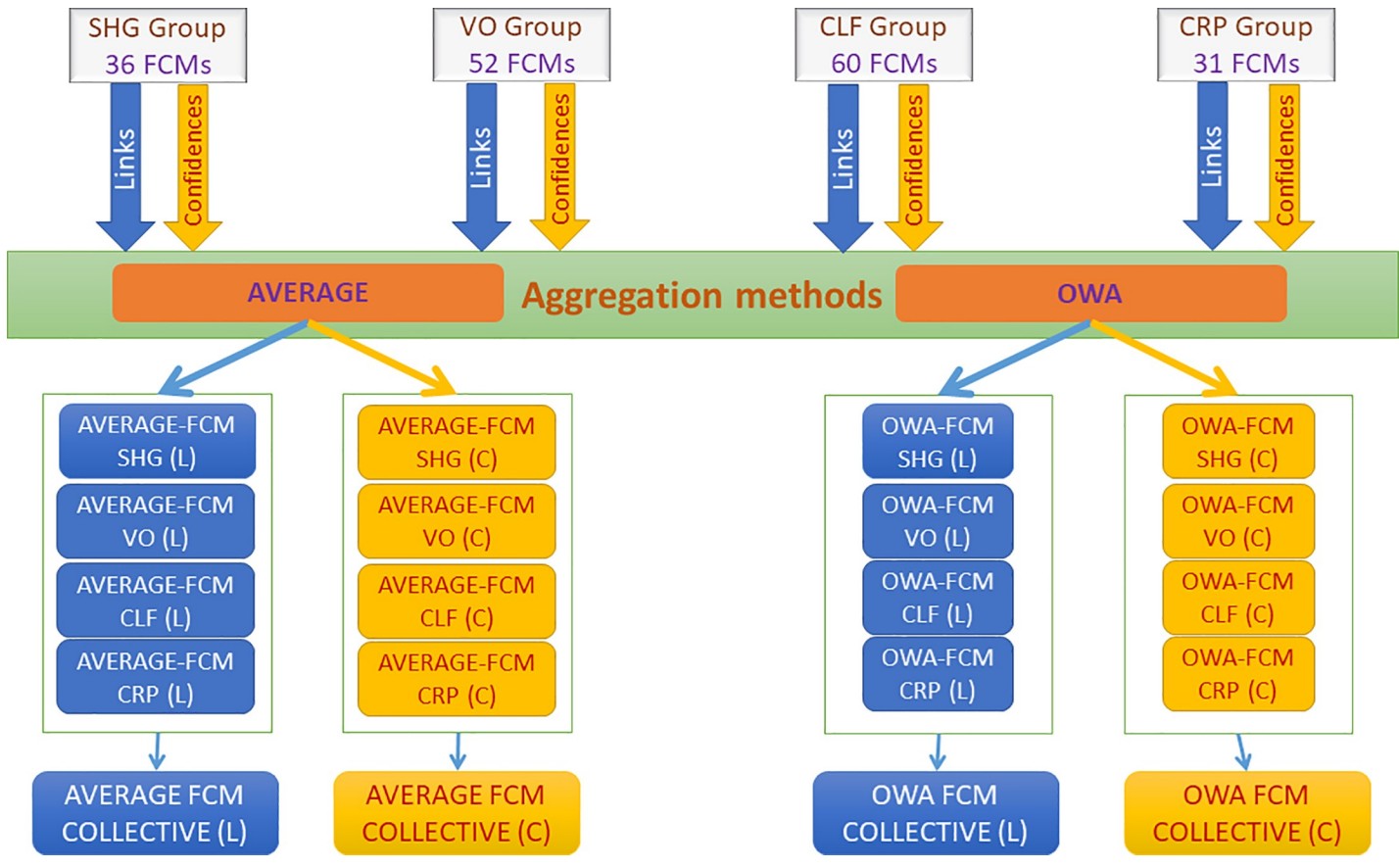

**Fig 4. The steps of the aggregation process.**

so that

$$\sum_i w_i = 1; \ w_i \in [0, 1] \tag{3}$$

and

$$f(a_1, \ldots, a_n) = \sum_{j=1}^{n} w_j \, b_j, \tag{4}$$

where, $b_j$ is the j$^{\text{th}}$ largest object of the collection of the aggregated elements $a_1, a_2 \ldots, a_n$. The function value $f(a_1, \ldots, a_n)$ determines the aggregated value of arguments $a_1, a_2 \ldots, a_n$.

Our calculations show that the re-ordering step is a basic characteristic of the OWA operator. In particular, an argument $a_i$ is not associated with a particular weight $w_i$ but rather a weight $w_i$ is associated with the ordered position $i$ of the arguments. An essential characteristic of the OWA operators is that, in order to select the vector $W$ properly, the *Max*, *Min* and arithmetic average operators need to be defined:

i. For $W = \begin{bmatrix} 1 \\ 0 \\ \vdots \\ 0 \end{bmatrix}, f(a_1, \ldots, a_n) = \text{Max}_i a_i$

ii. For $W = \begin{bmatrix} 0 \\ 0 \\ \vdots \\ 1 \end{bmatrix}, f(a_1, \ldots, a_n) = \text{Min}_i a_i$

iii. For $W = \begin{bmatrix} 1/n \\ 1/n \\ \vdots \\ 1/n \end{bmatrix}, f(a_1, \ldots, a_n) = \frac{1}{n}\sum_{i=1}^{n} a_i$

Brown [33] emphasises that the OWA operators include aggregation operators, inheriting the characteristics of the *Max* and *Min* operators regarding the commutatively, monotonicity, and idempotency:

$$\text{Min}_i a_i \leq f(a_1, \ldots, a_n) \leq \text{Max}_i a_i \qquad (5)$$

As this category of operators runs between the *Max* (*or*) and the *Min* (*and*), Yager [14] introduced the measurement function to define the type of the aggregation performed concerning a particular value, namely the *orness measure* of the aggregation, which is calculated by Eq (6):

$$Orness\ (W) = \frac{1}{n-1}\sum_{i=1}^{n}(n-1)w_i \qquad (6)$$

As suggested by Yager [14], this measure determines whether the aggregation works as an *or* (*Max*) operation; following equations describe it:

$$orness\ ([1\ 0 \ldots 0]^T) = 1, \qquad (7)$$

$$orness\ ([0\ 0 \ldots 1]^T) = 0, \qquad (8)$$

$$orness\ \left(\left[\frac{1}{n}\frac{1}{n} \ldots \frac{1}{n}\right]^T\right) = 0.5. \qquad (9)$$

Consequently, the *Max*, *Min*, and arithmetic average operators are considered as OWA operators, taking a degree of *orness*, the values 1, 0, and 0.5, respectively.

Yager [14] introduced a second measure, as dispersion or entropy connected with a weighting vector.

$$Disp\ (W) = \sum_{i=1}^{n} w_i \ln w_i \qquad (10)$$

The degree of information used, based on *W*, throughout the aggregation process, may be calculated by the above equation.

O'Hagan [34] generated the OWA weights, which have a predefined degree of *orness a* and maximise the entropy, with a method that uses these elements. He called them MEOWA

operators. The following constrained optimisation problem defines this procedure:

$$Maximize \sum_{i=1}^{n} w_i \ln w_i \tag{11}$$

$$subject\ to\ a = \frac{1}{n-1} \sum_{i=1}^{n} (n-1)w_i \tag{12}$$

$$\sum_{i=1}^{n} w_i = 1,\ w_i \in [0,1], i = (1, \ldots, n). \tag{13}$$

We observe that we need to specify the desired degree of *orness a*. Papageorgiou et al. present the FCM-OWA tool used for aggregating FCMs [19, 20].

**2.4.3. Learning OWA operators for aggregating FCM weights.** In this research, we present a new algorithm that may be used to aggregate the weights assigned by the experts and stakeholders during the FCMs' designing process. The proposed algorithm learns the weights linked with a specific use of the OWA operator from the experts and stakeholders. We can obtain the OWA weights through the following process:

Initially, experts' opinions are considered argument values $(a_{k1}, a_{k2}\ldots,a_{kn})$.

- Step 1: Create a slightly different parameter $\rho$ for each argument that indicates the optimism of the decision-maker, $0 \leq \rho \leq 1$.

- Step 2: Compute the aggregated values for each sample with the Hurwics method, where the aggregated value $d$ obtained from a tuple of $n$ arguments, $a_1, a_2,\ldots,a_n$, is defined as a weighted average of the *Max* and *Min* values of that tuple.

$$\rho\ \underset{i}{\text{Max}}\ a_i + (1 - \rho) \underset{i}{\text{Min}}\ a_i = d \tag{14}$$

- Step 3: Reorder the objects $a_{k1}, a_{k2}\ldots,a_{kn}$.

- Step 4: Compute the current estimation of the aggregated values $d_k$

$$\hat{d}_k = b_{k1}w_1 + b_{k2}w_2 + \cdots + b_{kn}w_n \tag{15}$$

through initial values of the OWA weights $w_1 = 1/n$.

- Step 5: Calculate the total $\hat{d}_k$, $d_k$, $b_{ki}$ for each $i$. The parameters $\lambda_i$ determine the weights of OWA and are updated with the propagation of the error $\hat{d}_k - d_k$ between the current estimated aggregated value and the actual aggregated value (see Eq (15)).

- Step 6: Compute the current estimations of the $\lambda_i$

$$\lambda_i(1+1) = \lambda_i(1) - \beta w_i(1)(b_{ki} - \hat{d}_k)(\hat{d}_k - d_k) \tag{16}$$

through initial values $\lambda_i(0) = 0$, $i = (1,\ldots,n)$, and a learning rate of $\beta = 0.35$.

- Step 7: Use $\lambda_i$, $i = (1,\ldots,n)$, for providing the latest estimate of the weights

$$w_i = \frac{e^{\lambda_i(1)}}{\sum_{j=1}^{n} e^{\lambda_i(1)}}, i = (1, n) \tag{17}$$

- Step 8: Update $w_i$ and $\hat{d}_k$ at each iteration, until the estimations for all the $\lambda_i$ converge to, that is $\Delta = |\lambda(l+1) - \lambda(l)|$ are small.

## 2.5. Groups aggregation for producing an overall collective FCM

Next, two Collective-FCMs are produced from each one of the four groups (SHG-FCM, VO-FCM, CLF-FCM, and CRP-FCM) using the weighted average aggregation method as well as the OWA-based aggregation method. Thus, a Collective-FCM is produced from the implementation of each aggregation method. The Collective-FCM is enriched with the knowledge of all stakeholders involved.

The four different Collective-FCM models considering links (L) and confidences and links (C), for both methods, are shown in Fig 5A, 5B, 5C and 5D.

## 3. Fuzzy cognitive map-based analysis and simulations

Typically, an FCM of $n$ concepts could be represented mathematically by an $n \times n$ weight matrix ($W$). By feeding the fuzzy cognitive map with an initial stimulus state vector $X^{(t)}$ (state vector at the time ($t$)), the progress of the scenarios can be modelled over time by evolving forward and letting concepts interact with one another. Each subsequent value of the concept state $X^{(t+1)}$ can be computed as previous state $X^{(t)}$ and weight matrix multiplication, according to Eq (18).

$$X_i^{(t+1)} = f\left( \sum_{\substack{j=1 \\ j \neq i}}^{n} w_{ji} \times X_j^t \right) \tag{18}$$

Based on the literature, two other equations have been proposed for FCM inference, the modified Kosko (Eq (19)) and the rescaled Kosko (Eq (20)).

$$X_i^{(t+1)} = f\left( X_i(t) + \sum_{\substack{j=1 \\ j \neq i}}^{n} X_j(t) \cdot w_{ji} \right) \tag{19}$$

$$X_i^{(t+1)} = f\left( (2 \times X_i^t - 1) + \sum_{\substack{j=1 \\ j \neq i}}^{n} w_{ji} \times (2 \times X_j^t - 1) \right) \tag{20}$$

where, $X_i^{(t+1)}$ is the value of the concept $C_i$ at simulation and step $t+1$, $X_j^{(t)}$ is the value of the concept $C_j$ at the simulation step $t$; $w_{ji}$ is the weight of the interconnection between concept $C_j$ and concept $C_i$, and $f(\cdot)$ is the threshold transfer function used to retain the values within the range [0,1] or [-1,+1]. In general, the most commonly used transfer function is Sigmoid, as shown in the Eq (21) [35].

$$f(x) = \frac{1}{1 + e^{-\lambda x}} \tag{21}$$

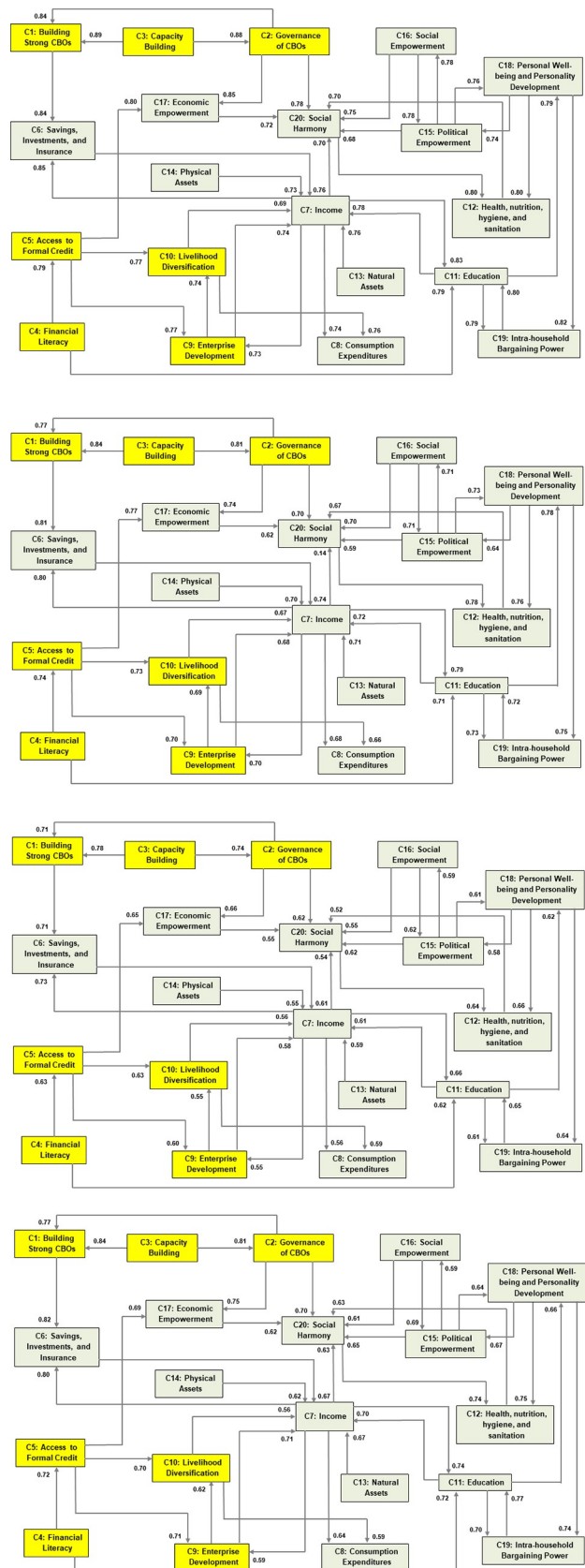

**Fig 5.** (a) Collective-FCM model for Average-FCM (L). (b) Collective-FCM model for OWA-FCM (L). (c) Collective-FCM model for Average-FCM (C). (d) Collective-FCM model for OWA-FCM (C).

where $\lambda$ is a real positive number ($\lambda > 0$) which defines the steepness of the continuous function $f$ while $x$ is the value $X_i^{(t)}$ for a given iteration.

The simulation stops when the system reaches equilibrium, i.e., a limit vector is reached as $X^t = X^{t-1}$ or when $X^t - X^{t-1} \leq e$; where $e$ is a residual, describing the minimum error difference among the subsequent concepts and typically has a value of 0.001.

## 3.1. Structural analysis

The structural analysis provides a description of FCM models with the help of indices including in-degree (weight of inbound links), out-degree (weight of outbound links), degree centrality (sum of the corresponding absolute weights of in-degree and out-degree), complexity, density, and the hierarchy index [28, 36]. We conducted the structural analysis aided by the FCMWizard tool. The values calculated, regarding the degree centrality, complexity, density and hierarchy index are 3.07, 0.25, 0.103 and 0.035 respectively, for the overall OWA aggregated FCM. These structural indices help in analysing the graphical structure of FCMs [21, 28, 32].

## 3.2. Development of input vectors for policy scenarios

As discussed in the theory of change in Section 1.1, the institutional effectiveness of CBOs is critical for effective operation and for achieving the end goal of the programme. Thus, in order to understand the effectiveness of CBOs, we selected the following concepts of the FCM model from the four elements of the programme for FCM-based simulations: C1-"Building strong CBOs", C2-"Governance of CBOs", C3-"Capacity Building", C5-"Access to formal credit", C9-"Enterprise development", and C10-"Livelihood diversification". These concepts were selected as they describe the four elements of the programme, while being among the concepts with the highest degree centrality (see section 4.1) so that they could well influence the dynamics of the system. The scenarios selected with their input vector concepts are briefly presented in Table 1.

- *Scenario 1* examines the effects of building strong CBOs (C1) and the governance of CBOs (C2) in terms of SHGs, VO, and CLFs, while *Scenario 2* presents the effects of capacity building of the CBOs (C3) in terms of governance and management.

- *Scenario 3* studies the effects of access to formal credit (C5) in terms of micro-finance and SHG-bank linkage.

- *Scenario 4* highlights the effects of enterprise development of the CBOs (C9) in terms of business sustainability, along with livelihood diversification (C10) in terms of living standards.

- The rest of the scenarios examined are combinations of the above four scenarios. In particular, *Scenario 5* shows the effects of building strong CBOs (C1) along with the governance of CBOs (C2) and the capacity building of the CBOs (C3).

- *Scenario 6* illustrates the combination of the effects of building strong CBOs (C1) in terms of SHGs, VOs, and CLFs, the governance of CBOs (C2), and the access to formal credit (C5).

- *Scenario 7* considers the effects of building strong CBOs (C1) and the governance of CBOs (C2) along with enterprise development (C9) and livelihood diversification (C10) of the CBOs.

- *Scenario 8* examines the effects of access to formal credit (C5), enterprise development (C9) and livelihood diversification (C10) of the CBOs.

**Table 1. The input vector concepts of each scenario.**

| Scenarios | Input vector concepts | | | | | |
|---|---|---|---|---|---|---|
| *Scenario 1* | C1: Building strong CBOs | C2: Governance of CBOs | | | | |
| *Scenario 2* | C3: Capacity building | | | | | |
| *Scenario 3* | C5: Access to formal credit | | | | | |
| *Scenario 4* | C9: Enterprise development | C10: Livelihood diversification | | | | |
| *Scenario 5* | C1: Building strong CBOs | C2: Governance of CBOs | C3: Capacity building | | | |
| *Scenario 6* | C1: Building strong CBOs | C2: Governance of CBOs | C5: Access to formal credit | | | |
| *Scenario 7* | C1: Building strong CBOs | C2: Governance of CBOs | C9: Enterprise development | C10: Livelihood diversification | | |
| *Scenario 8* | C5: Access to formal credit | C9: Enterprise development | C10: Livelihood diversification | | | |
| *Scenario 9* | C1: Building strong CBOs | C2: Governance of CBOs | C3: Capacity building | C5: Access to formal credit | C9: Enterprise development | C10: Livelihood diversification |

- *Scenario 9* highlights the effects of all the pre-selected vital concepts: building strong CBOs (C1), governance of CBOs (C2), capacity building of the CBOs (C3), access to formal credit (C5), enterprise development (C9), and livelihood diversification (C10) of the CBOs.

The simulation results of the above scenarios help us to understand the critical factors to achieve the desired programme outcomes.

### 3.3. Simulation process

The FCM-based simulations were carried out using the FCMWizard tool [21]. Each concept in the FCM model has a state variable that varies from |0| to |1| and is associated with an activation variable. In other words |0| means 'non-activated' and |1| means 'activated' [29, 31, 37, 38]. When one or more concepts are 'activated' by inputting an initial non-zero value, this activation spreads through the matrix following the weighted relationships. Each iteration produces a new state vector with 'activated' and 'non-activated' concepts. Feedback loops cause repeated activation of a concept, introducing non-linearity to the model [9, 38, 39]. The activation of concepts is iterated, using a 'squashing function' to rescale concept values towards |1|, until the model reaches the equilibrium or the steady-state [38]. The simulation process culminates with the attainment of a steady-state of the system. To determine the steady-state, we ran a simulation process starting with an initial state vector $X_0$, with the input vectors identified in each scenario (1 to 9) clamped to 1. An equilibrium or steady-state vector is obtained after the FCM convergence [9, 29, 37, 38]. The resulting concept values can be used to interpret the outcomes of a particular scenario and to study the dynamics of the modelled system [29–31].

We applied the modified Kosko activation rule (Eq (19)) proposed by Stylios and Groumpos [8] to run simulations, because of its memory capabilities, along with the Sigmoid transfer function (Eq (21)) as the concepts only have positive values.

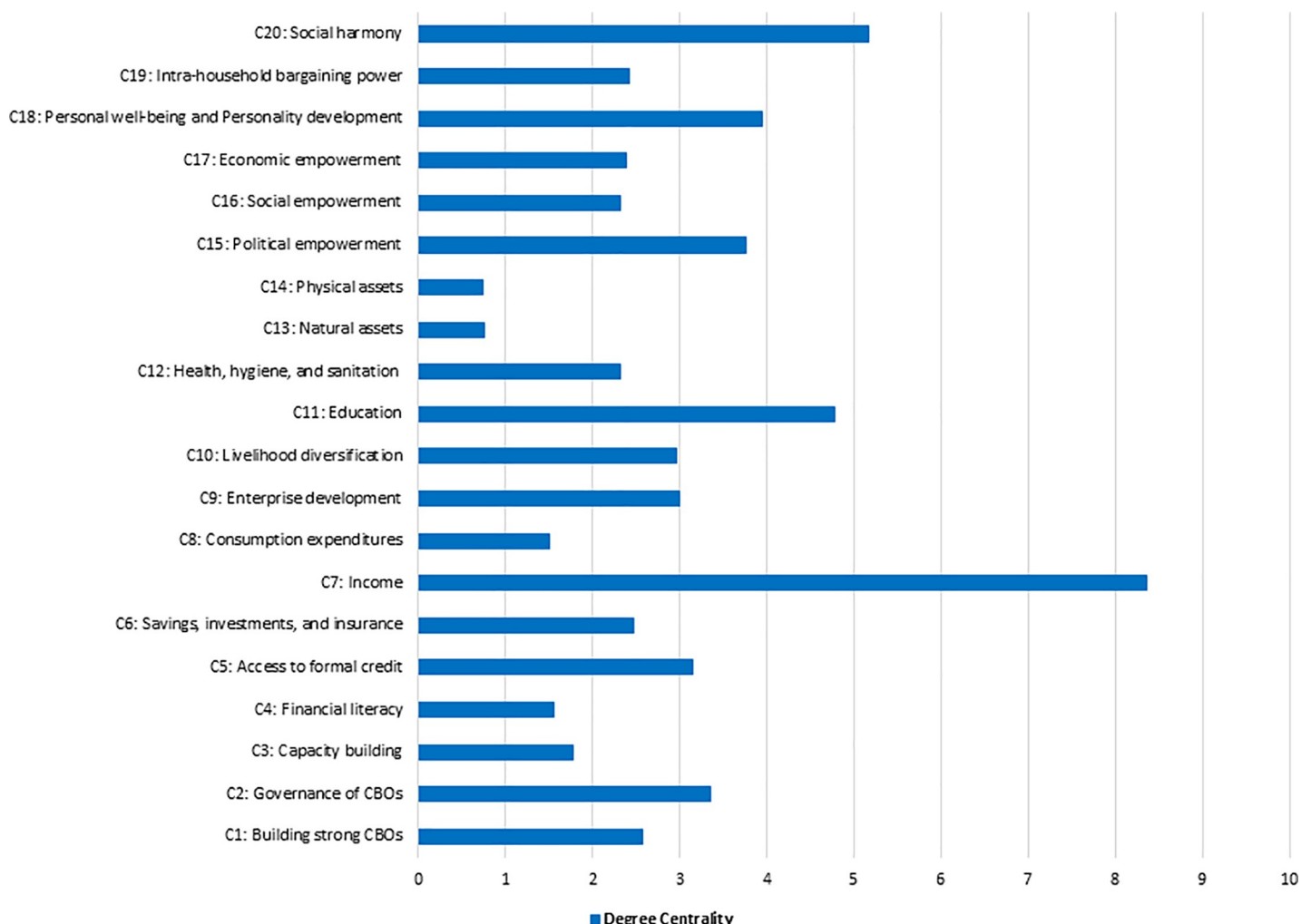

**Fig 6. Finalised concepts and their centrality for the Collective Average-FCM (L).**

## 4. Results and discussions

### 4.1. Characteristics of key concepts

It is necessary to limit the concepts to the most influential and uncertain while dealing with a large number of concepts, which are challenging to analyse individually. Some independent concepts are disconnected from the system, while some dependent concepts have a relatively low degree of influence but strong dependence. Filtering key concepts from the FCM system is a traditional approach in scenario planning that helps link narratives to the quantitative model while focussing on the critical concepts with strong direct or indirect effects on the scenario objectives, which can significantly change the balance of the entire system [40]. In the FCM-based scenario analysis, the identification of key concepts mainly relies on the perception of the experts. However, some characteristics obtained from the model facilitate the process.

The weights of the links reveal three useful structural indices for this matter: in-degree, out-degree, and degree centrality [28, 36, 41]. Degree centrality is considered as the relative importance of a concept within the FCM structure [36]. The calculated values of degree centrality for the Collective Average-FCM (L), along with the concepts previously, are summarised in Fig 6.

## 4.2. Aggregation results

This section illustrates the results produced from the application of the two aggregation methods on the FCM models. An example being the matrices containing the calculated values of weights for the two Collective-FCMs with confidences and links (C), the Average-FCM (C) and the OWA-FCM (C), after the application of the Average and OWA aggregation processes respectively. The corresponding results of the other two Collective-FCMs considering links (L), the Average-FCM (L) and the OWA-FCM (L), are illustrated in S2 and S3 Figs.

In order to proceed with the comparison analysis, mean deviations among the aggregated Average-FCM and OWA-FCM, for both links (L) and confidences and links (C), has been calculated. The results are shown in Table 2. Before that, the deviations between the weights among all the above FCMs (in pairs) have also been calculated.

When evaluating the values produced from the results presented in Table 3, it is observed that the minimum deviation value is located between the OWA-FCM with confidence and links (C) and the Experts' FCM, while, the next smallest value lies in between Average-FCM with confidence and links (C) and the Experts' FCM. It may be concluded that the OWA-FCM model resembles the structure of the Expert-based FCM and can, consequently, present a similar performance to the model constructed by the experts. Therefore, the superiority of the OWA-FCM model against the Average-FCM may be inferred after an adequate evaluation and interpretation of the results produced.

## 4.3. Scenarios results

This section details out the results of the FCM-based simulations where scenario analysis is conducted. Simulating the FCM model provides a profound understanding of the concepts' behaviour, their relationships, and the magnitude of impacts on other concepts. We performed the scenario analysis by multiplying the input vectors with the adjacency matrix while applying the Sigmoid threshold function with $\lambda = 1$ after every multiplication. The process was iterated until the system (output vectors) reached the steady-state.

Before the simulation process takes place, a baseline scenario needs to be conducted. The next step will show a comparison between the results of specific scenarios and those resulting from the state of the model where all the initial values of concepts are zero (baseline scenario). In S4 and S5 Figs, the results of simulations of the baseline scenarios for both aggregated methods, considering the FCM models with links (Average-FCM (L) and OWA-FCM (L)) are presented.

Subsequently, the simulation process is performed by "clamping" the initial values of the input vector concepts one-by-one (see Eq (19)). The results are compared against the baseline scenario where the system reaches the steady-state by clamping all the initial values to zero. Exploring the dynamic change of the values of the concepts between the baseline steady-state and the results of the clamping procedure enables quantitative interpretation of the impact of the key concepts on the system.

**Table 2. Mean deviations among the aggregated FCMs.**

| Comparing FCMs | | Mean Deviation |
|---|---|---|
| OWA-FCM (L) | Average-FCM (L) | 0.32 |
| OWA-FCM (C) | Average-FCM (C) | 0.37 |
| Experts' FCM | Average-FCM (L) | 0.27 |
| Experts' FCM | Average-FCM (C) | 0.24 |
| Experts' FCM | OWA-FCM (C) | 0.12 |

**Table 3. Scenario results (initial and final value) for each concept, for the Expert-based FCM.**

| Key Concept | Scenario 1 | | Scenario 2 | | Scenario 3 | | Scenario 4 | | Scenario 5 | | Scenario 6 | | Scenario 7 | | Scenario 8 | | Scenario 9 | |
|---|---|---|---|---|---|---|---|---|---|---|---|---|---|---|---|---|---|---|
| | Initial value | Final value | Initial value | Final value | Initial value | Final value | Initial value | Final value | Initial value | Final value | Initial value | Final value | Initial value | Final value | Initial value | Final value | Initial value | Final value |
| C1 | 1 | 1 | 0 | 0.906 | 0 | 0.874 | 0 | 0.874 | 1 | 1 | 1 | 1 | 1 | 1 | 0 | 0.874 | 1 | 1 |
| C2 | 1 | 1 | 0 | 0.827 | 0 | 0.780 | 0 | 0.780 | 1 | 1 | 1 | 1 | 1 | 1 | 0 | 0.780 | 1 | 1 |
| C3 | 0 | 0.659 | 1 | 1 | 0 | 0.659 | 0 | 0.659 | 1 | 1 | 0 | 0.659 | 0 | 0.659 | 0 | 0.659 | 1 | 1 |
| C4 | 0 | 0.659 | 0 | 0.659 | 0 | 0.659 | 0 | 0.659 | 0 | 0.659 | 0 | 0.659 | 0 | 0.659 | 0 | 0.659 | 0 | 0.659 |
| C5 | 0 | 0.765 | 0 | 0.765 | 1 | 1 | 0 | 0.765 | 0 | 0.765 | 1 | 1 | 0 | 0.765 | 1 | 1 | 1 | 1 |
| C6 | 0 | 0.910 | 0 | 0.904 | 0 | 0.902 | 0 | 0.903 | 0 | 0.910 | 0 | 0.910 | 0 | 0.911 | 0 | 0.903 | 0 | 0.911 |
| C7 | 0 | 0.962 | 0 | 0.962 | 0 | 0.963 | 0 | 0.968 | 0 | 0.962 | 0 | 0.963 | 0 | 0.968 | 0 | 0.968 | 0 | 0.968 |
| C8 | 0 | 0.790 | 0 | 0.790 | 0 | 0.790 | 0 | 0.791 | 0 | 0.790 | 0 | 0.790 | 0 | 0.791 | 0 | 0.791 | 0 | 0.791 |
| C9 | 0 | 0.864 | 0 | 0.864 | 0 | 0.882 | 1 | 1 | 0 | 0.864 | 0 | 0.882 | 1 | 1 | 1 | 1 | 1 | 1 |
| C10 | 0 | 0.860 | 0 | 0.860 | 0 | 0.880 | 1 | 1 | 0 | 0.860 | 0 | 0.880 | 1 | 1 | 1 | 1 | 1 | 1 |
| C11 | 0 | 0.890 | 0 | 0.890 | 0 | 0.890 | 0 | 0.891 | 0 | 0.890 | 0 | 0.890 | 0 | 0.891 | 0 | 0.891 | 0 | 0.891 |
| C12 | 0 | 0.798 | 0 | 0.798 | 0 | 0.798 | 0 | 0.798 | 0 | 0.798 | 0 | 0.798 | 0 | 0.798 | 0 | 0.798 | 0 | 0.798 |
| C13 | 0 | 0.659 | 0 | 0.659 | 0 | 0.659 | 0 | 0.659 | 0 | 0.659 | 0 | 0.659 | 0 | 0.659 | 0 | 0.659 | 0 | 0.659 |
| C14 | 0 | 0.659 | 0 | 0.659 | 0 | 0.659 | 0 | 0.659 | 0 | 0.659 | 0 | 0.659 | 0 | 0.659 | 0 | 0.659 | 0 | 0.659 |
| C15 | 0 | 0.865 | 0 | 0.865 | 0 | 0.865 | 0 | 0.865 | 0 | 0.865 | 0 | 0.865 | 0 | 0.865 | 0 | 0.865 | 0 | 0.865 |
| C16 | 0 | 0.785 | 0 | 0.785 | 0 | 0.785 | 0 | 0.785 | 0 | 0.785 | 0 | 0.785 | 0 | 0.785 | 0 | 0.785 | 0 | 0.785 |
| C17 | 0 | 0.885 | 0 | 0.871 | 0 | 0.886 | 0 | 0.867 | 0 | 0.885 | 0 | 0.901 | 0 | 0.885 | 0 | 0.886 | 0 | 0.901 |
| C18 | 0 | 0.876 | 0 | 0.876 | 0 | 0.876 | 0 | 0.876 | 0 | 0.876 | 0 | 0.876 | 0 | 0.876 | 0 | 0.876 | 0 | 0.876 |
| C19 | 0 | 0.879 | 0 | 0.879 | 0 | 0.879 | 0 | 0.879 | 0 | 0.879 | 0 | 0.879 | 0 | 0.879 | 0 | 0.879 | 0 | 0.879 |
| C20 | 0 | 0.953 | 0 | 0.948 | 0 | 0.947 | 0 | 0.946 | 0 | 0.953 | 0 | 0.954 | 0 | 0.953 | 0 | 0.947 | 0 | 0.954 |

Concerning the Expert-based FCM model, the results of the scenarios are shown in Table 3. The results of the scenarios for the other aggregated FCMs (the Average and the OWA, for links (L) and confidences and links (C)) are depicted in S1–S4 Tables.

The scenario analysis performs simulations for the 9 scenarios that were formed in Section 3.2. Scenario 1 is dedicated to increasing the concepts C1-"Building strong CBOs" and C2-"Governance of CBOs" by "clamping" them to one, while scenarios 2 and 3 increase the concepts C3-"Capacity Building" and C5-"Access to formal credit" by "clamping" them to one, respectively. Scenario 4 refers to the increase of the concepts C9-"Enterprise development" and C10-"Livelihood diversification" by "clamping" them to one.

In Scenario 5, the key concepts C1-"Building strong CBOs", C2-"Governance of CBOs", and C3-"Capacity Building" are all clamped to one. The same procedure is followed for Scenario 6 where the key concepts C1-"Building strong CBOs, C2-"Governance of CBOs", and C5-"Access to formal credit" are clamped to one, while Scenario 7 refers to the concepts C1-"Building strong CBOs, C2-"Governance of CBOs", C9-"Enterprise development", and C10-"Livelihood diversification" clamped to one. Scenario 8 considers "clamping" the key concepts C5-"Access to formal credit", C9-"Enterprise development", and C10-"Livelihood diversification" to one. In the end, Scenario 9 is conducted by "clamping" all the key concepts (C1-"Building strong CBOs", C2-"Governance of CBOs", C3-"Capacity Building", C5-"Access to formal credit", C9-"Enterprise development", and C10-"Livelihood diversification") to one.

The four aggregated FCMs, (Average-FCM (L), Average-FCM (C), OWA-FCM (L), and OWA-FCM (C)), have been exerted to test nine plausible scenarios as mentioned above. Thus, the scenarios developed have been generalised to the poverty alleviation programme of India, with the poverty alleviation change scenarios extracted.

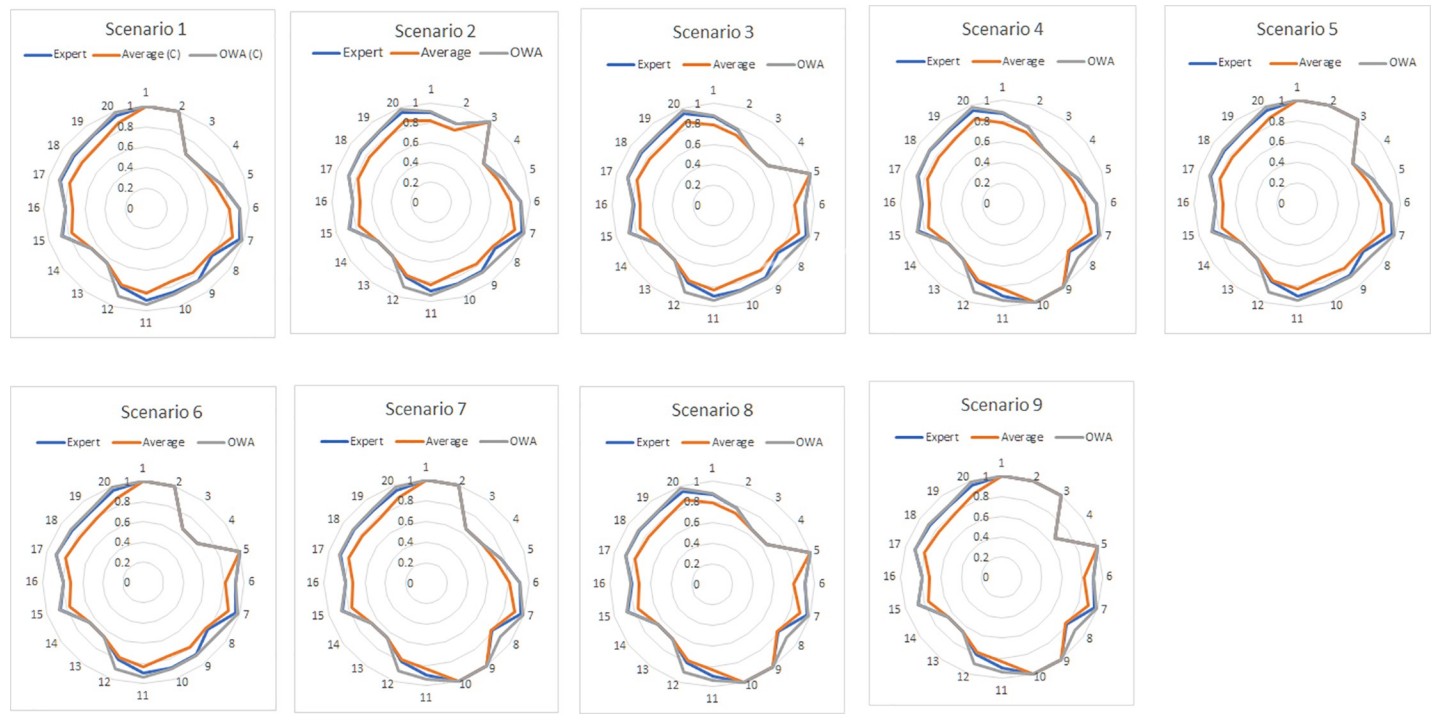

**Fig 7. Spider graphs for the Expert-based, Average, and OWA-FCMs (C).**

The results for scenario analysis are illustrated in Figs 7 and 8. In our case, we considered the Expert-based FCM as the benchmark model that helped us to further investigate the

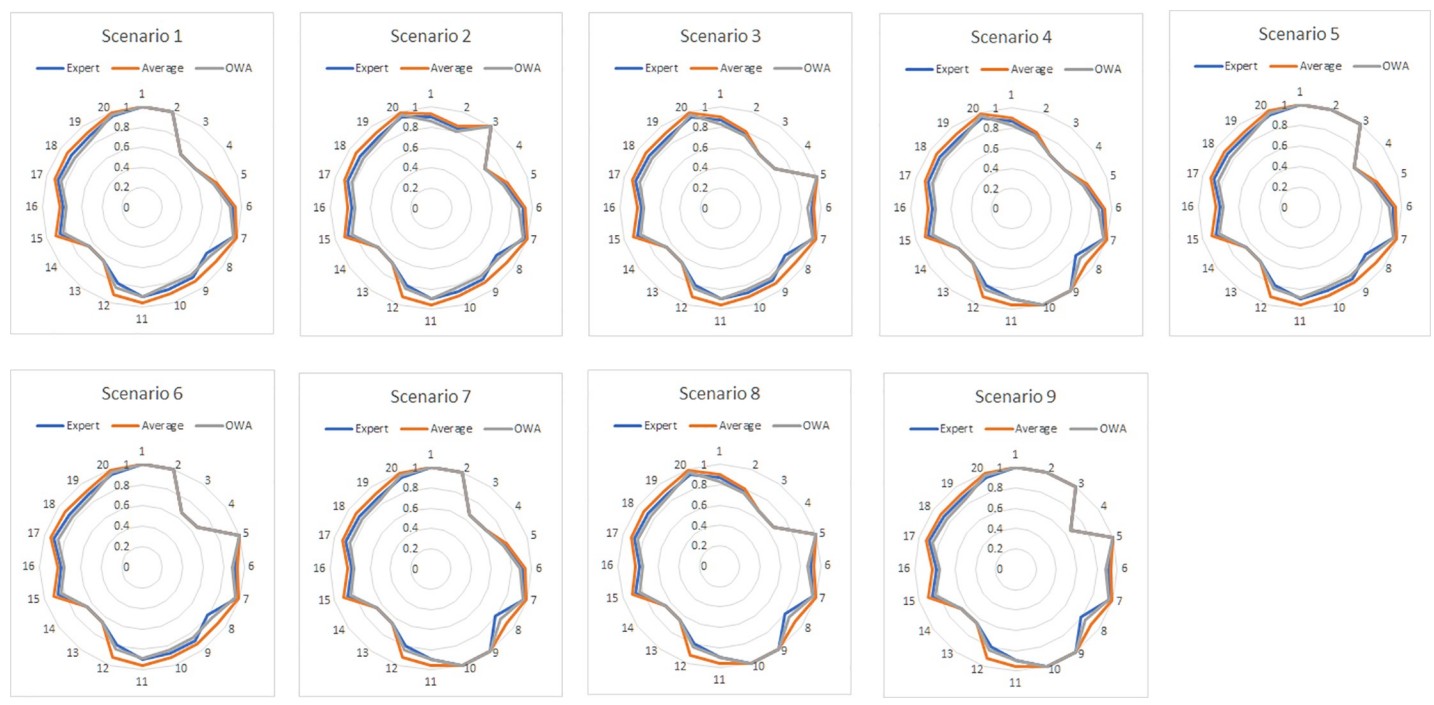

**Fig 8. Spider graphs for the Expert-based, Average, and OWA-FCMs (L).**

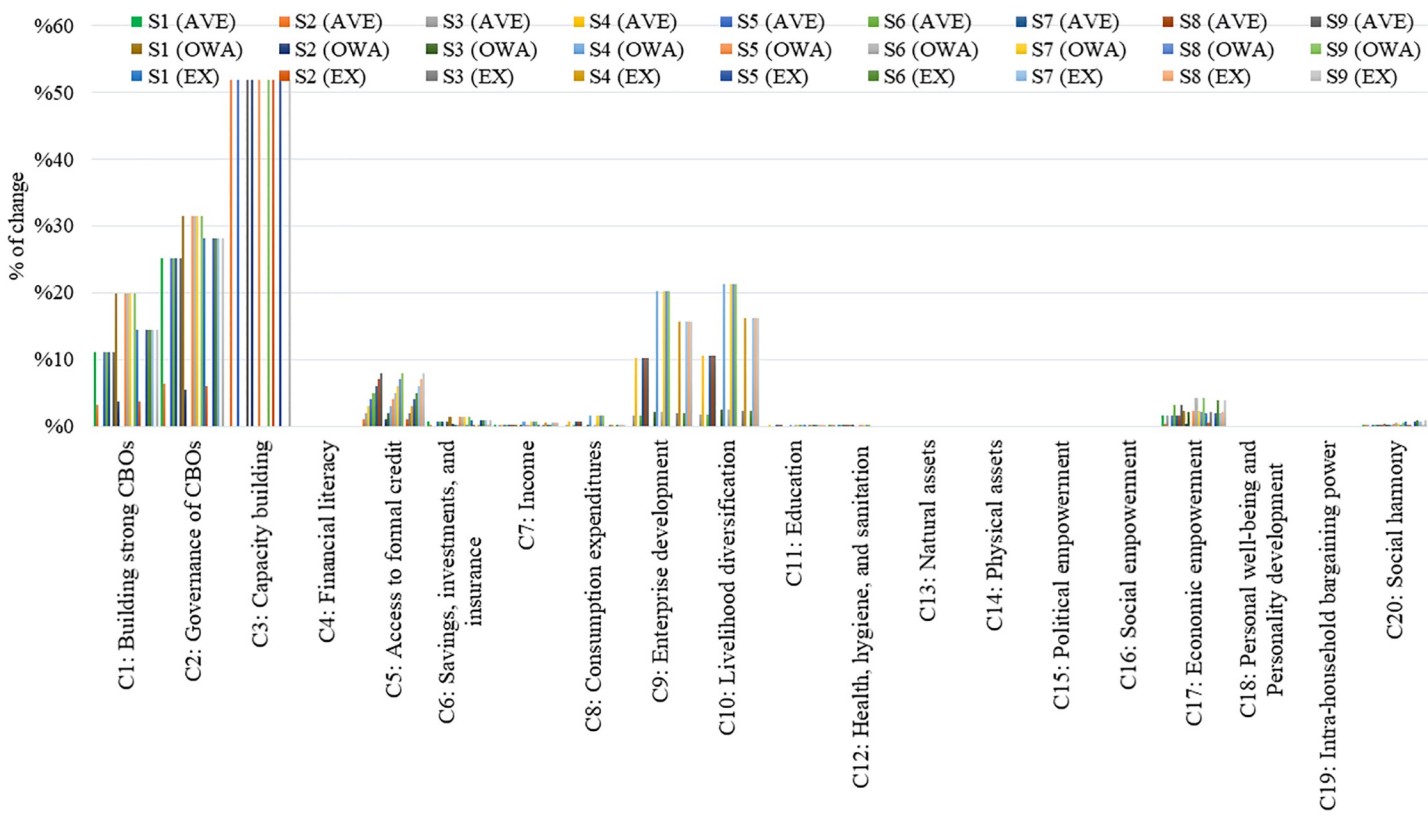

**Fig 9. Scenario analysis for all FCMs considering confidences and links.**

usefulness, importance, and superiority of the proposed OWA aggregation method against the Average aggregation method.

To begin with, Figs 7 and 8 illustrate spider charts for all three FCMs (Average, OWA, and Expert-based) and the deviation for all concepts after the nine scenarios were conducted while considering links (L) plus confidences and links (C) correspondingly.

All the scenarios examined presented in Figs 7 and 8, for both cases (for the links and for the confidences and links), reveal that the produced analysis results of the OWA FCM are in high consistency with those of the Expert-based FCM. The OWA method seems to resemble the Expert-based FCM in terms of performance as the chart lines of the OWA and the Expert-based FCMs coincide in most of the cases of the scenarios produced, as presented in Figs 7 and 8. The results produced work as indicators to verify the superiority of the OWA-FCM aggregation over the Average-FCM, making OWA a trustworthy method with regard to FCMs' aggregation.

Fig 9 that follows, gathers all the outcomes after the execution of the scenarios, with respect to the percentage of change for each concept, for all FCMs considering Confidences and Links (Average, OWA, and Expert-based).

Among the concepts identified in the final outcomes of the Theory of Change that could affect the end goal in terms of decreased socio-economic poverty, concepts C17 and C20 seem to have emerged as notable deviations from the initial states after the implementation of scenario analysis (see Fig 9). Therefore, the scenario analysis mainly focusses on the impact that the key concepts have on these two output concepts C17-"Economic Empowerment" and

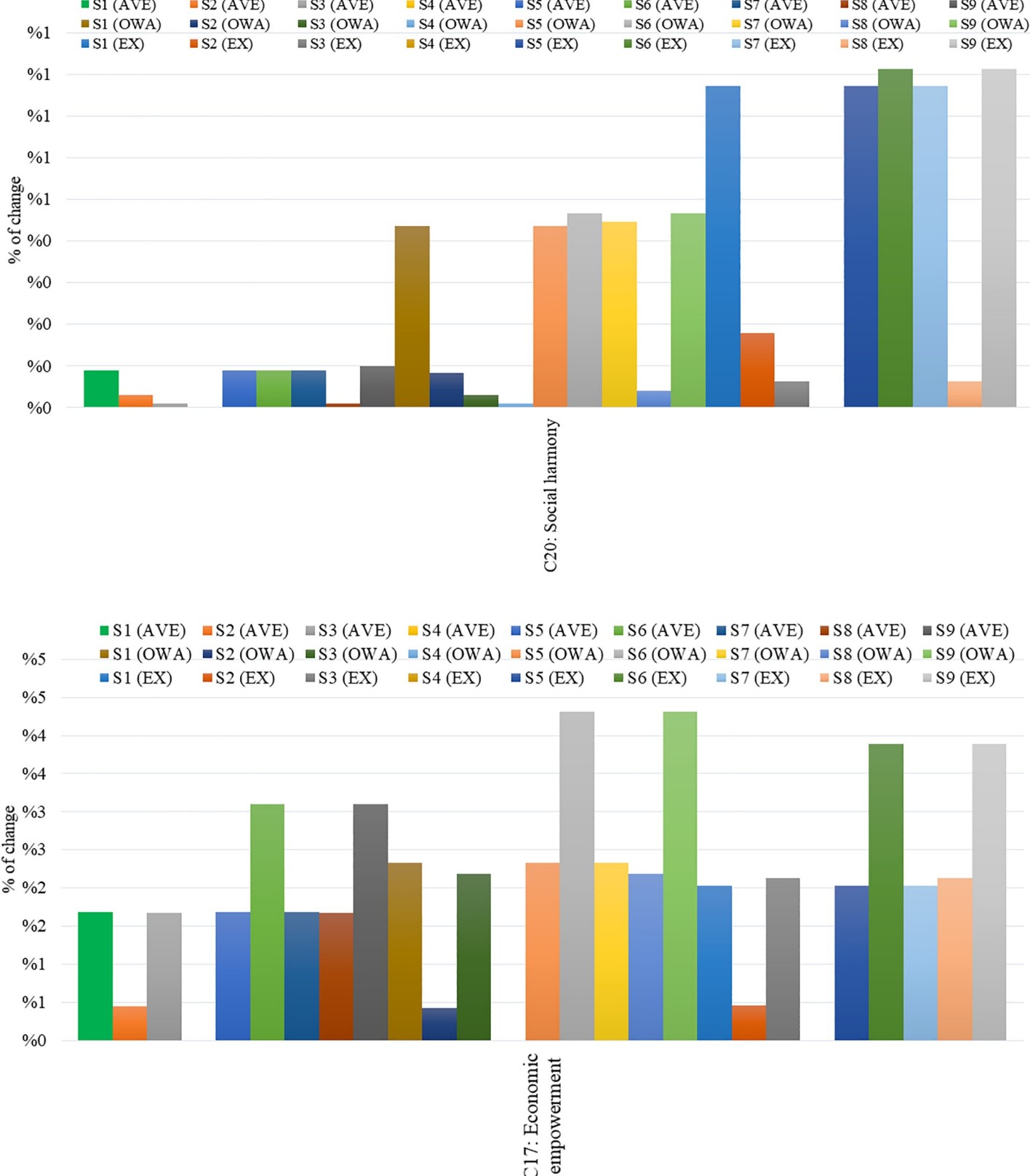

**Fig 10.** Deviations for concepts (a) C20 and (b) C17 for aggregated FCMs considering confidences and links.

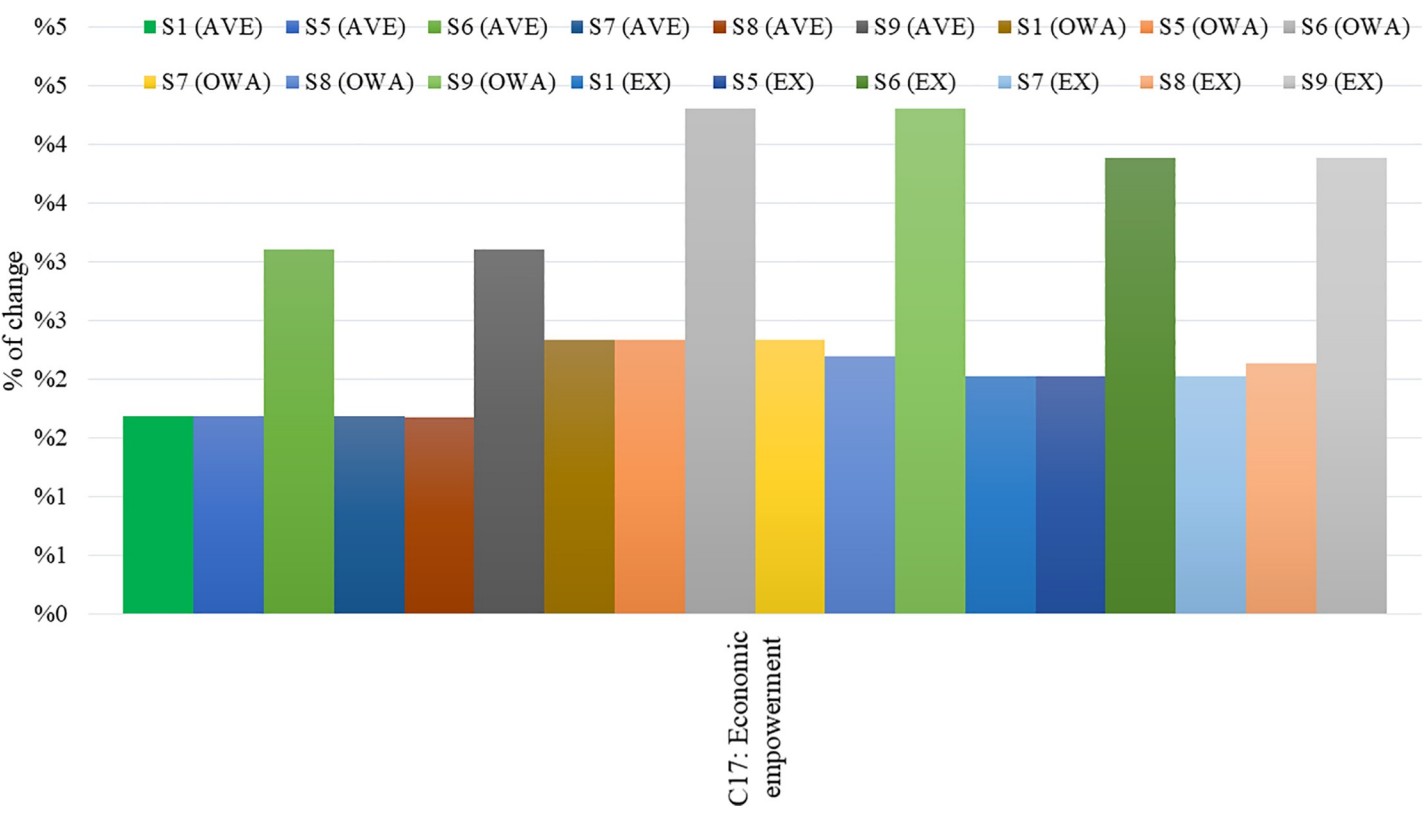

**Fig 11. Outcome concept C17 percentage of change, considering confidences and links.**

C20-"Social Harmony", which researchers consider as the outcome concepts that need to be improved by the DAY-NRLM programme interventions.

Thus, we calculate the deviations for these outcome concepts for all three FCMs, considering only confidences and links (C). The results for concepts C20 and C17 are depicted in Fig 10.

After having conducted a detailed analysis of the results presented above, it may be concluded that the percentage of change for both outcome concepts C20 and C17 is notable only for six out of the nine scenarios (S1, S5, S6, S7, S8, and S9) performed. Results of the scenarios S2, S3, and S4 seems to be less significant, on the other hand. Therefore, only these six scenarios are taken into consideration hereafter.

Fig 11 helps researchers in this study to conclude that out of the two applied aggregation methods, the Average and the OWA, the latter is more efficient with respect to the accuracy of the results compared to the Expert-based FCM model, which is considered a benchmark model by the researchers for the purposes of this study.

In Fig 11, the change in percentage of the outcome concept C17, compared to the initial steady-state, is presented for all three approaches (Average, OWA, and Expert-based) for confidences and links (C). The respective deviations calculated for the outcome concept C20, for all three FCMs, are depicted in Fig 12.

## 4.4. Policy implications of the scenario results

The scenarios 1 to 4, previously analysed, illustrate the importance of strong CBOs, good governance within CBOs, better capacity building of communities and CBOs, access to formal

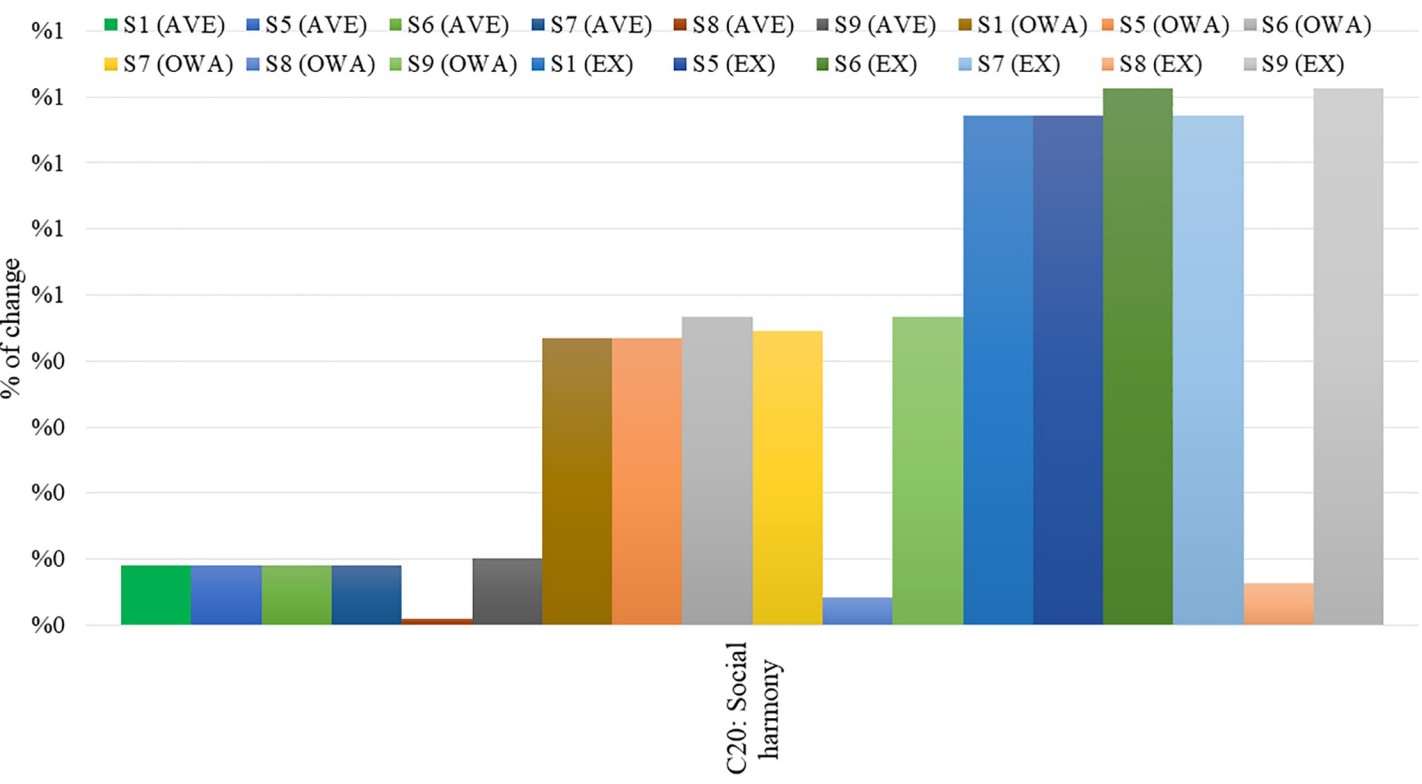

**Fig 12. Outcome concept C20 percentage of change, considering confidences and links.**

credit, livelihood diversification, and enterprise development to achieve the end goal of the programme, i.e. decrease in socio-economic poverty. The results show that if CBOs habitually mainstream financial institutions, offer support through SHG-bank linkages, help livelihood diversification, and develop livelihood enterprises then one can expect the following: better capacity building of communities and CBOs along with good governance within CBOs. This will help shape strong CBOs. Good governance reflects adherence to the *Panchsutra* (regular meetings, regular savings, regular inter-loaning, timely repayments, and up-to-date books of accounts). Better access to formal credit could lead to the provisioning of savings and investments to stakeholders. Insurance, alongside, is also likely to increase. Besides, an increase in natural and physical assets is likely to increase household incomes. A better income is likely to increase the expenditure on consumer goods, healthcare, and education. Increased income and savings could as well lead to increased livelihood diversification and enterprise development. Further, greater access to formal credit and good governance within the CBOs is likely to result in economic, political, and social empowerment of SHG women besides improving social harmony within the community. Better education is likely to develop the personality and personal well-being of SHG women besides improving their intra-household bargaining power along with health, hygiene, and sanitation.

The scenarios 5 to 9 show different combinations of input vector concepts used in previous scenarios. The outcomes show a result similar to the previous scenarios. The building of CBOs and providing them with access to formal credit will probably enhance the economic, social, and political empowerment of women. Also, increasing income and savings could lead to higher consumer spending, diversification of livelihoods, and development of livelihood enterprises. Higher incomes could lead to a better access to education for women and their children,

helping develop their personality and personal well-being while improving their overall socio-economic status. Better education is also likely to improve their intra-household bargaining power and health, hygiene, and sanitation.

In general, the results confirm that several concepts are complementary and should be implemented simultaneously for the overall development of SHG members. It is necessary to improve the capacities of SHG members while ensuring good governance within the CBOs and providing micro-finance through high-quality CBOs. Access to micro-finance and higher income will help them to diversify their livelihood options and develop micro-enterprises. This will result in socio-economic empowerment of women, provide them with social safety nets while improving their education, health and hygiene and developing their personalities. Thus, the findings reveal that the creation of strong community-based institutions through capacity building and access to formal credits will help the alleviation of socio-economic poverty.

## 5. Discussion

The dimensions of poverty eradication in the rural area of India present a complex system. Fig 10 reveals that the concepts of the scenarios S2, S3, S4 with respect to all three FCM approaches (Average, OWA, and Expert-based) for confidences and links (C), have a small impact on the final outcome C17 (Economic empowerment) and C20 (Social Harmony) since the percentage of change from the initial steady-state for the concepts C17 and C20 seems to be insignificant when the concepts of the scenarios implemented are "clamped" to one. For this reason, the researchers in this study ignored these scenarios and focussed on the rest of scenarios (S1, S5, S6, S7, S8, and S9) to further investigate their impact on poverty alleviation.

From Figs 11 and 12, it emerges that the combination of key concepts C1, C2, C3, C5, C9, and C10 (Scenario 9) has the highest impact on key concepts C17 and C20 for all aggregated FCMs, showing a significant increase in these concepts, particularly when OWA aggregation method is applied, considering confidences and links. With respect to the performance of the two aggregation methods examined, the following key remarks are drawn after a careful investigation of the above Tables and Figs:

i.  In the participatory modelling, when a large number of participants are involved, the OWA aggregation method appears superior to the Average aggregation method. After a detailed observation of Figs 11 and 12, where the scenario analysis of the three different FCMs, considering confidences and links (C), is conducted, it is being observed that deviations from the initial state are high when the OWA aggregation method is applied. In both cases, the OWA method performs better in the decision-making process, compared to the Average aggregation method.

ii. The proposed OWA aggregation method exhibits outstanding performance while performing scenario analysis for the concepts examined. This may emanate from the fact that the results produced by the application of the OWA aggregation method (when referring to the FCMs either with links (L) or confidences and links (C) are closer to those deriving from the Expert-based FCM model, which is considered the benchmark model. Therefore, they surpass those of the Average aggregation method (Figs 11 and 12). The group of experts, as stated above, can create an accurate model that could well describe the problem examined in order to make appropriate strategic decisions.

iii. On the other hand, in all the cases examined, the Average aggregation method appears less important in decision-making when its outcomes are matched with the FCM model created by the experts.

iv. The proposed OWA aggregation method exhibits better performance compared to the Expert-based, considering the case of the FCMs with confidences and links (C) with regard to the key concepts C17 and C20. As can be seen in Figs 9, 11 and 12, results derived from the application of the OWA aggregation method are better than those from the experts.

v. The performance of the OWA aggregated FCM model is similar to that of the Expert-based FCM model in terms of the consistency of results produced between the two approaches (Figs 11 and 12).

In general, it is validated that the OWA aggregation method outperforms the Average aggregation method and is suitable for scenario analysis, strategic decision-making, and policy-planning, especially when it involves a large number of participants.

## 6. Conclusions

This study examined the contribution of the OWA-based FCM aggregation method by learning the weights of the OWA operator from the data for aggregation tasks. We applied this methodology to the study of the DAY-NRLM poverty alleviation programme, which aims at socio-economic growth, economic sustainability, and livelihood diversification of poor women in rural areas in India. Using a combination of interventions—building strong CBOs, good governance within CBOs, better capacity building of communities and CBOs, access to formal credit to the community, livelihood diversification, and enterprise development—extreme poverty can be reduced.

What makes this study significant is the relationship strengths that were calculated with the proposed aggregation approach and compared with the benchmark weights (average) along with those assigned by the experts. The results show that the overall performance of the OWA-FCM mimics that of the Expert-based FCM. The scenarios carried out based on different FCMs try to model the circumstances relevant to improving livelihoods by creating sustainable and self-managed institutional platforms, together with the promotion of social resilience and economic empowerment of the poor rural women. A significant and evident outcome that makes this study important, concerning the scenario analysis results, is that it reveals an overall OWA-based FCM vis-à-vis the Expert-based FCM and the Average FCM. The study also presents similar trends regarding the influence of certain key concepts such as women's political, social, and economic empowerment and other critical key concepts, including increasing social harmony within the community. The coherence between the impacts of certain key concepts of the OWA-FCM model and the corresponding expert-based concepts has been demonstrated very well in the study. This coherence makes the proposed method significant for policymakers, outperforming the Average aggregation method.

Moreover, decision-makers can apply the proposed aggregation method along with the new software tools for policy-making in various domains, proving its generic applicability and convenience when a significantly large number of participants are involved in designing FCMs.

## Supporting information

**S1 Fig. The FCM protocol used for data collection from the participants.**
(TIF)

**S2 Fig. Adjacency weights matrix for Average-FCM (L) model.**
(TIF)

**S3 Fig. Adjacency weights matrix for OWA-FCM (L) model.**
(TIF)

**S4 Fig. The baseline scenario for overall Average-FCM (L) model.**
(TIF)

**S5 Fig. The baseline scenario for overall OWA-FCM (L) model.**
(TIF)

**S1 Table. Scenario results (initial and final value) for each concept, for the Average-FCM (L).**
(DOCX)

**S2 Table. Scenario results (initial and final value) for each concept, for the Average-FCM (C).**
(DOCX)

**S3 Table. Scenario results (initial and final value) for each concept, for the OWA-FCM (L).**
(DOCX)

**S4 Table. Scenario results (initial and final value) for each concept, for the OWA-FCM (C).**
(DOCX)

## Acknowledgments

This work was supported by the independent research framework of the Institute for Bio-economy and Agri-technology, Center for Research and Technology Hellas, Greece. We sincerely thank the participants who took part in the research. We thank the Academic Editor and the reviewers for their insightful comments.

## Author Contributions

**Conceptualization:** Konstantinos Papageorgiou.

**Data curation:** Pramod K. Singh, Harpalsinh Chudasama.

**Formal analysis:** Konstantinos Papageorgiou, Elpiniki I. Papageorgiou.

**Investigation:** Konstantinos Papageorgiou, Pramod K. Singh, Elpiniki I. Papageorgiou, Harpalsinh Chudasama.

**Methodology:** Konstantinos Papageorgiou, Elpiniki I. Papageorgiou.

**Project administration:** Elpiniki I. Papageorgiou.

**Resources:** Pramod K. Singh.

**Software:** Konstantinos Papageorgiou.

**Supervision:** Elpiniki I. Papageorgiou, George Stamoulis.

**Validation:** Pramod K. Singh, Harpalsinh Chudasama.

**Visualization:** Konstantinos Papageorgiou.

**Writing – original draft:** Konstantinos Papageorgiou, Elpiniki I. Papageorgiou.

**Writing – review & editing:** Pramod K. Singh, Elpiniki I. Papageorgiou, Dionysios Bochtis, George Stamoulis.

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
