## [Decision Letter · Decision Letter 0]

7 Feb 2020

PONE-D-19-35078

Participatory modelling for poverty alleviation using fuzzy cognitive maps and OWA learning aggregation

PLOS ONE

Dear Mr. Papageorgiou,

Thank you for submitting your manuscript to PLOS ONE. After careful consideration, we feel that it has merit but does not fully meet PLOS ONE’s publication criteria as it currently stands. Therefore, we invite you to submit a revised version of the manuscript that addresses the points raised during the review process.

Though Reviewer 1 rejected PONE-D-19-35078, the reviewer provided many valuable and constructive comments. Considering reviewers’ useful comments and the interesting topic of the manuscript, I would like to give you a chance to revise your manuscript. The revised manuscript will undergo the next round of review by the same reviewers.

We would appreciate receiving your revised manuscript by Mar 23 2020 11:59PM. To enhance the reproducibility of your results, we recommend that if applicable you deposit your laboratory protocols in protocols.io, where a protocol can be assigned its own identifier (DOI) such that it can be cited independently in the future. For instructions see: http://journals.plos.org/plosone/s/submission-guidelines#loc-laboratory-protocols

We look forward to receiving your revised manuscript.

Kind regards,

Baogui Xin, Ph.D.

Academic Editor

PLOS ONE

Journal Requirements:

2. We note that Figures 1, 7 and 8 in your submission contain copyrighted images. All PLOS content is published under the Creative Commons Attribution License (CC BY 4.0), which means that the manuscript, images, and Supporting Information files will be freely available online, and any third party is permitted to access, download, copy, distribute, and use these materials in any way, even commercially, with proper attribution. For more information, see our copyright guidelines: http://journals.plos.org/plosone/s/licenses-and-copyright.

1.         You may seek permission from the original copyright holder of Figures 1, 7 and 8 to publish the content specifically under the CC BY 4.0 license.

"Part of this work has received funding from the Ministry of Rural Development, Government

of India for providing financial support."

4. Please ensure that you refer to Figure 2 in your text as, if accepted, production will need this reference to link the reader to the figure.

Reviewers' comments:

Reviewer's Responses to Questions

**Comments to the Author**

1. Is the manuscript technically sound, and do the data support the conclusions?

Reviewer #1: Partly

Reviewer #2: Yes

Reviewer #3: Yes

2. Has the statistical analysis been performed appropriately and rigorously? 

Reviewer #1: I Don't Know

Reviewer #2: Yes

Reviewer #3: Yes

3. Have the authors made all data underlying the findings in their manuscript fully available?

Reviewer #1: Yes

Reviewer #2: Yes

Reviewer #3: Yes

4. Is the manuscript presented in an intelligible fashion and written in standard English?

Reviewer #1: Yes

Reviewer #2: Yes

Reviewer #3: Yes

5. Review Comments to the Author

Reviewer #1: --------------------------------------------------------------------------------------------------------------------------------------------------------------------------------------------------------

Reviewer #2: The article title is appropriate

The abstract accurately reflects the content.

The primary objective is genuine and argued persuasively

The purported significance of the article is explicitly stated.

The article adequately ties to the relevant literature.

The research study methods are sound and appropriate.

The writing is clear, concise and interesting however further proofreading should be done to correct some typographical mistakes. Also the author should check the Sentence in line numbers 253-255 (i.e. Each adjacency matrix consists of the normalised ……………………was called FCM (L) model) and correct the grammatical error therein.

All figures, tables, and images are necessary and appropriate.

The conclusions are accurate and supported by the content.

The article has made sufficient contribution to knowledge

Not all articles in reference are cited. Authos should delete references not cited in the manuscript.

Reviewer #3: The paper is well written but there are some points:

-Figures are not clear at all.

-It is recommended to compare this proposed model with new developed FCMs such Neuro-FCM (NFCM).

-It is expected to proof the performance of proposed model in mathematical manner not just giving an example.

6. PLOS authors have the option to publish the peer review history of their article (what does this mean?). If published, this will include your full peer review and any attached files.

Reviewer #1: No

Reviewer #2: Yes: Boluwaji Ade Akinnuwesi

Reviewer #3: No

---

## [Author Response · Author response to Decision Letter 0]

29 Feb 2020

Comments by the Academic Editor 

1 Please include the following items when submitting your revised manuscript: 

• A rebuttal letter that responds to each point raised by the academic editor and reviewer(s). This letter should be uploaded as separate file and labeled 'Response to Reviewers'.

• A marked-up copy of your manuscript that highlights changes made to the original version. This file should be uploaded as separate file and labeled 'Revised Manuscript with Track Changes'.

• An unmarked version of your revised paper without tracked changes. This file should be uploaded as separate file and labeled 'Manuscript'. 

Responses of Authors: The authors have included all the required items with the revised manuscript.

2 Please ensure that your manuscript meets PLOS ONE's style requirements, including those for file naming. The PLOS ONE style templates can be found at

 The authors have ensured that the manuscript meets PLOS ONE's style requirements, including those for file naming.

3 We note that Figures 1, 7 and 8 in your submission contain copyrighted images. All PLOS content is published under the Creative Commons Attribution License (CC BY 4.0), which means that the manuscript, images, and Supporting Information files will be freely available online, and any third party is permitted to access, download, copy, distribute, and use these materials in any way, even commercially, with proper attribution. For more information, see our copyright guidelines: http://journals.plos.org/plosone/s/licenses-and-copyright.

I. You may seek permission from the original copyright holder of Figures 1, 7 and 8 to publish the content specifically under the CC BY 4.0 license.

II. If you are unable to obtain permission from the original copyright holder to publish these figures under the CC BY 4.0 license or if the copyright holder’s requirements are incompatible with the CC BY 4.0 license, please either i) remove the figure or ii) supply a replacement figure that complies with the CC BY 4.0 license. Please check copyright information on all replacement figures and update the figure caption with source information. If applicable, please specify in the figure caption text when a figure is similar but not identical to the original image and is therefore for illustrative purposes only. 

Responses of Authors: Figures 7 and 8 have been removed in the revised manuscript. They are outcomes of a java-based tool that has been developed by PhD student, for research purposes and no software license has been attributed yet. 

In Figure 1, we had inserted SDG1 graphics, which is removed in the new Figure in the revised manuscript. Since we earlier worked on the national evaluation of DAY-NRLM, the Theory of Change for DAY-NRLM, which we had developed in consultation with the programme implementers, had some pictorial designs. We developed the Theory of Change as depicted in Figure 1 of this paper in consultation with the programme implementers (Jammu and Kashmir State Rural Livelihood Mission). All the pictorial graphics are removed in the current version of Figure 1. This Figure is similar but not identical to the one developed for the national DAY-NRLM.

4 Thank you for stating the following in the Acknowledgments Section of your manuscript:

"Part of this work has received funding from the Ministry of Rural Development, Government of India for providing financial support."

Responses of Authors: Funding related text is deleted from the manuscript. The Funding Disclosure Statement is updated. 

5 Please ensure that you refer to Figure 2 in your text as, if accepted, production will need this reference to link the reader to the figure. 

Responses of Authors: The authors have provided the reference to Figure 2 in the text in the revised manuscript.

Comments by Reviewer #1 and Authors’ Responses

1 

(No comments) • 

Responses of Authors: We thank the reviewer for approving the manuscript for publication fully. 

Reviewer #2:

Comments by Reviewer #2 and Authors’ Responses

1 The article title is appropriate.

The abstract accurately reflects the content.

The primary objective is genuine and argued persuasively.

The purported significance of the article is explicitly stated.

The article adequately ties to the relevant literature.

The research study methods are sound and appropriate.

The writing is clear, concise and interesting however further proofreading should be done to correct some typographical mistakes. 

Responses of Authors: We thank you very much for your appreciation and insightful comments, which helped improve the paper. We have tried to address all your concerns. 

2 Also, the author should check the Sentence in line numbers 253-255 (i.e. Each adjacency matrix consists of the normalised………………was called FCM (L) model) and correct the grammatical error therein. 

Responses of Authors: This is corrected in the revised version of the manuscript.

3 All figures, tables, and images are necessary and appropriate.

The conclusions are accurate and supported by the content.

The article has made sufficient contribution to knowledge

Not all articles in reference are cited. Authors should delete references not cited in the manuscript. 

Responses of Authors: The list of references have been updated and correctly cited in the revised version of the manuscript.

Reviewer #3:

Comments by Reviewer #3 and Authors’ Responses

1 The paper is well written but there are some points:

 Responses of Authors: We welcome and appreciate your insightful comments, which helped improving the paper. We have tried to address the points raised in the revised manuscript. 

2 Figures are not clear at all. 

Responses of Authors: All the Figures in this study are created in the best possible resolution and were uploaded individually according to the journal’s guidelines. However, during the initial submission, the Figures were somehow distorted after being zoomed in and transformed to pdf which was automatically created by the system.

3 It is recommended to compare this proposed model with new developed FCMs such Neuro-FCM (NFCM). 

Responses of Authors:

 • In response to the Reviewer’s comment, the authors added this reference in the introduction section, with a brief discussion about NFCM and its capabilities in decision-making tasks.

• In particular, for the construction of the NFCM, the weights of connecting links are determined by using a neuro-fuzzy system (NFIS). NFIS approach needs empirical data in the form of fuzzy rules to define the weights of the model. By using the known weight values, the network is trained until the desired values of NFIS parameters are obtained.

• The most important issue in designing an NFCM model, as described in Amirkhani, et al. (2019), is the construction of an appropriate set of fuzzy rules defined by experts for this model and the calculation of model weights through training. 

• Thus, it is evident that NFCM uses empirical data to build a neuro-fuzzy system for modeling the inherent fuzziness that exists in them. 

• The authors of this study didn't deal with empirical data (fuzzy rules) as needed for NFCM construction and presented in the work of Amirkhani, et al. (2019). Instead, we followed a specific protocol using stakeholders’ perceptions to design the FCM model by assigning numerical values to weighted interconnections among concepts for policy-making purposes in the domain of socio-economic poverty alleviation of poor rural communities. 

• Thus, it is not feasible to provide a straightforward comparison with NFCM, and we are stack to this common FCM design process. A simple reference though, on the NFCM and its applicability under certain conditions, is desired.

4 It is expected to proof the performance of proposed model in mathematical manner not just giving an example. 

Responses of Authors: 

• Authors do not need to prove the implemented approach mathematically as this is exclusively based on a previous approach, which has been proven and presented in Papageorgiou, et al. (2020) and dealt with the OWA aggregation method and especially for learning OWA operators. Specifically, in that work, a step by step application of the learning OWA operators for defining the OWA FCM weights was presented, in a simple FCM that was constructed by the experts. 

• The proposed model was accordingly adapted to the context of this study and particularly for the aggregation of FCM weights. Thus, there was no need to present it further and again, mathematically prove its performance.

• For the convenience of the Reviewers, the authors cite their previous work in “Sustainability”, where a complete example is presented Papageorgiou, et al. (2020).

---

## [Decision Letter · Decision Letter 1]

30 Mar 2020

PONE-D-19-35078R1

Participatory modelling for poverty alleviation using fuzzy cognitive maps and OWA learning aggregation

PLOS ONE

Dear Mr. Papageorgiou,

Thank you for submitting your manuscript to PLOS ONE. After careful consideration, we feel that it has merit but does not fully meet PLOS ONE’s publication criteria as it currently stands. Therefore, we invite you to submit a revised version of the manuscript that addresses the points raised during the review process.

Since the first reviewer is not satisfied with your revision, I give you the last chance to revise your manuscript. To speed the review process, the revised manuscript will only be reviewed by the Academic Editor in the next round.

We would appreciate receiving your revised manuscript by May 14 2020 11:59PM. To enhance the reproducibility of your results, we recommend that if applicable you deposit your laboratory protocols in protocols.io, where a protocol can be assigned its own identifier (DOI) such that it can be cited independently in the future. For instructions see: http://journals.plos.org/plosone/s/submission-guidelines#loc-laboratory-protocols

We look forward to receiving your revised manuscript.

Kind regards,

Baogui Xin, Ph.D.

Academic Editor

PLOS ONE

Reviewers' comments:

Reviewer's Responses to Questions

**Comments to the Author**

1. If the authors have adequately addressed your comments raised in a previous round of review and you feel that this manuscript is now acceptable for publication, you may indicate that here to bypass the “Comments to the Author” section, enter your conflict of interest statement in the “Confidential to Editor” section, and submit your "Accept" recommendation.

Reviewer #1: (No Response)

Reviewer #2: All comments have been addressed

Reviewer #3: All comments have been addressed

2. Is the manuscript technically sound, and do the data support the conclusions?

Reviewer #1: Yes

Reviewer #2: Yes

Reviewer #3: Yes

3. Has the statistical analysis been performed appropriately and rigorously? 

Reviewer #1: Yes

Reviewer #2: Yes

Reviewer #3: I Don't Know

4. Have the authors made all data underlying the findings in their manuscript fully available?

Reviewer #1: Yes

Reviewer #2: No

Reviewer #3: (No Response)

5. Is the manuscript presented in an intelligible fashion and written in standard English?

Reviewer #1: Yes

Reviewer #2: Yes

Reviewer #3: Yes

6. Review Comments to the Author

Reviewer #1: The manuscript Overall is well structured and well organised. It also contributes to the emerging FCM-based scenario domains. However, I have some minor and a few major concerns about some parts of the context.

I understand that paragraph four (A new extension of FCM, known as NFCM … experts and the calculation of model weights through training.) was later provided in the revision version in response to a comment by a reviewer. However, it appears entirely disengaged from the main topic while adding no value to the text. I would suggest moving this paragraph to the ‘Discussion’ section, which can be further expanded by analysing the similarities, differences, advantages and possible disadvantages of each technique. This modification will be in turn in line with the ‘Discussion’ context.

Section one (1. Introduction) is simply too long. The subsection 1.1 (Theory of change for decreased socio-economic poverty in Jammu and Kashmir) while being too long in the title, can come separately and independently as a case study section, perhaps after the methodology section. This subsection indeed comes between two main parts of the section, the first introducing the research, and the second providing the contributions of the paper along with the summarisation of the paper structure.

In section 2.2.1 (Expert-based FCM using an open-concept design) who are those 31 experts (their job title, position, their field, etc.)? How many workshops were arranged?

Line 360, I guess the correct number for the figure caption “Fig 3a. Collective-FCM model for Average-FCM (L)” is Fig. 5a.

Sub-titles are often too long. When possible, please shorten by keeping only the keywords.

In the text, ‘Figure’ sometimes come as abbreviated ‘Fig’, and sometimes come fully ‘Figure’. The label should be consistent through the whole text. The order of figures at the end is pretty sloppy. It starts with ‘Fig. 2’-‘Fig. 14’ and then ends with ‘Fig. 1’ and ‘Fig. 3’.

Section 2 is again too long and contains arguably too many subsections. Breaking down this section into two sections could improve the structure of the work.

Figures 7&8 can be moved to the supplementary information as they do not offer many contributions to the richness of the text body.

Discussion can come as an independent section, and it is better to come after ‘Policy implications of the scenario results’ to keep the coherency.

In section 2.8 (Structural analysis), the paper promises providing indices such as complexity, density, and the hierarchy index, but they were never brought.

Reviewer #2: PONE-D-19-35078R1

Participatory modelling for poverty alleviation using fuzzy cognitive maps and OWA learning aggregation

PLOS ONE

COMMENTS

1. Recommendation (Accept, Minor Revision, Major Revision, Reject) – Accept

2. Is the manuscript technically sound, and do the data support the conclusions? (Answer options: Yes, No, Partly) - Yes

3. Has the statistical analysis been performed appropriately and rigorously? (Answer options: Yes, No, I don't know, N/A) - Yes

4. Does the manuscript adhere to the PLOS Data Policy? Additional details can be found at http://www.plosone.org/static/policies#sharing - Yes

5. Is the manuscript presented in an intelligible fashion and written in standard English? (Answer options: Yes, No) - Yes

6. Review Comments to the Author - Below are my comments

The abstract is good. The primary objective is good.

Introduction: This is good and comprehensive. It aligns with the research objective. However, it is too long. I advise the author to reduce the length of the introduction. The key issues should be focused on.

The Literature reviewed is okay.

Methodology: This is good and well directed towards achieving the aim and objective of the research. However, I advise the author to reduce the length and hence focus on only the most important concepts adopted and the model developed and the implementation.

The figures and graphs are okay.

The conclusion is okay.

General Comments: The manuscript is too long. The author should focus only on key concepts relevant to the model development and implementation with view to reduce the length of the manuscript.

7. Would you like your identity revealed to the authors of this submission? (Answer options: Yes, No) - No

8. Do you have any potentially competing interests? If none, type "None." Our policy on competing interests can be found at http://www.plosone.org/static/policies.action#competing. - None

Reviewer #3: It is well written and all the comments are addressed. Now it is ready for publication..............

7. PLOS authors have the option to publish the peer review history of their article (what does this mean?). If published, this will include your full peer review and any attached files.

Reviewer #1: No

Reviewer #2: No

Reviewer #3: Yes: Abdollah Amirkhani

---

## [Author Response · Author response to Decision Letter 1]

15 May 2020

Reviewer #1:

1 The manuscript Overall is well structured and well organised.

It also contributes to the emerging FCM-based scenario domains. However, I have some minor and a few major concerns about some parts of the context.

I understand that paragraph four (A new extension of FCM, known as NFCM … experts and the calculation of model weights through training.) was later provided in the revision version in response to a comment by a reviewer.

However, it appears entirely disengaged from the main topic while adding no value to the text. I would suggest moving this paragraph to the

‘Discussion’ section, which can be further expanded by analysing the similarities, differences, advantages and possible disadvantages of each technique. This modification will be in turn in line with the

‘Discussion’ context. 

Authors’ Response 1.1

• We thank you very much for your appreciation and insightful comments, which helped improve the paper. We have tried to address all your concerns.

• Thank you for the comment concerning this paragraph. We have significantly shortened it, keeping only the most important key issues. Thus, we believe that this paragraph will be well engaged with the content of this section and does not need to be transferred to another section.

• Also, we reduced the length of certain sections, like the “introduction” and “methodology”, shortening the overall length of the revised manuscript, according to Reviewers’ concerns, without omitting the most significant issues.

2 Section one (1. Introduction) is simply too long. The subsection 1.1 (Theory of change for decreased socio-economic poverty in Jammu and Kashmir) while being too long in the title, can come separately and independently as a case study section, perhaps after the methodology section. This subsection indeed comes between two main parts of the section, the first introducing the research, and the second providing the contributions of the paper along with the summarisation of the paper structure. 

Authors’ Response 1.2

The sub-title is shortened. 

The placing of this sub-section is appropriate as the theory of change helps us to develop input vectors for policy scenarios. In methodology section, the rationale for deciding input vectors will not be clear unless it is discussed before the methodology section. Instead of keeping under methodology section, we prefer to place it before.

3 In section 2.2.1 (Expert-based FCM using an open-concept design) who are those 31 experts (their job title, position, their field, etc.)? How many workshops were arranged? 

Authors’ Response 1.3

During expert-based FCM using an open-concept design we had one workshop participated by 31 national and state-level implementers of the DAY-NRLM programme. The participants constituted 6 generalists (in-charge of administration of the programme) and 22 specialists in the area of community development, micro-finance, farm and non-farm livelihoods, and monitoring and evaluation including three experts who were involved in designing the programme. We classified all of them as implementers of the programme.

4 Line 360, I guess the correct number for the figure caption “Fig 3a. Collective-FCM model for Average-FCM (L)” is Fig. 5a. Sub-titles are often too long. When possible, please shorten by keeping only the keywords. 

Authors’ Response 1.4

• In the previously revised manuscript, the figure caption was correctly written as Fig 5a and not as Fig 3a.

• Figures captions have been shortened, where needed.

5 In the text, ‘Figure’ sometimes come as abbreviated ‘Fig’, and sometimes come fully ‘Figure’. The label should be consistent through the whole text. The order of figures at the end is pretty sloppy. It starts with ‘Fig. 2’-‘Fig. 14’ and then ends with ‘Fig. 1’ and ‘Fig. 3’. 

Authors’ Response 1.5

• The word “Figure” was replaced by the abbreviation “Fig” throughout the revised manuscript, following your suggestion.

• Authors checked carefully the order of all figures in the manuscript and amended or deleted the parts that possibly caused confusion to readers.

6 Section 2 is again too long and contains arguably too many subsections. Breaking down this section into two sections could improve the structure of the work. 

Authors’ Response 1.6

• Thank you for your comment. Section 2 has been split into two new sections. Section 2 of the revised manuscript presents the methodology for FCM development, whereas section 3 contains the scenario analysis and simulation process.

7 Figures 7&8 can be moved to the supplementary information as they do not offer many contributions to the richness of the text body. Discussion can come as an independent section, and it is better to come after ‘Policy implications of the scenario results’ to keep the coherency.

Authors’ Response 1.7

• Thank you for the comment. Figures 7 and 8 have been moved to the supplementary information as S4 and S5 Figs.

• Following your suggestion, we also transferred the “Discussion” section after “Policy implications of the scenario results”.

8 In section 2.8 (Structural analysis), the paper promises providing indices such as complexity, density, and the hierarchy index, but they were never brought. 

Authors’ Response 1.8

• In the revised version of the manuscript, we added the values of these structural analysis indices (L351-L353), as requested by the Reviewer.

Reviewer #2:

1 The abstract is good. The primary objective is good. 

Introduction: This is good and comprehensive. It aligns with the research objective. However, it is too long. I advise the author to reduce the length of the introduction. The key issues should be focused on. 

The Literature reviewed is okay. 

Authors’ Response 2.1

• We thank you very much for your appreciation and insightful comments, which helped improve the paper. We have tried to address all your concerns.

• Authors revised the “Introduction” section by reducing its length, while trying to keep the key issues.

2 Methodology: This is good and well directed towards achieving the aim and objective of the research. However, I advise the author to reduce the length and hence focus on only the most important concepts adopted and the model developed and the implementation.

The figures and graphs are okay.

The conclusion is okay. 

Authors’ Response 2.2

• Thank you for the comment. Methodology (section 2) has been broken down into two main sections, as it was too long and difficult to follow. In the revised manuscript, section 2 presents the methodology for FCM development, whereas section 3 contains the scenario analysis and simulation process.

3 General Comments: The manuscript is too long. The author should focus only on key concepts relevant to the model development and implementation with view to reduce the length of the manuscript. 

Authors’ Response 2.3

• Thank you for the comment. The length of the manuscript has been reduced adequately to meet Reviewer’s concerns, keeping only the most important key issues. More specifically, we reduced the length of certain sections, like the “introduction” and “methodology”, shortening the overall length of the revised manuscript. The version with the track-changes will provide a clearer view of the changes that have been made.

Reviewer #3:

1 It is well written and all the comments are addressed. Now it is ready for publication. 

Authors’ Response 3.1

• We thank the reviewer for approving the manuscript for publication.

---

## [Editor Report · Decision Letter 2]

18 May 2020

Participatory modelling for poverty alleviation using fuzzy cognitive maps and OWA learning aggregation

PONE-D-19-35078R2

Dear Dr. Papageorgiou,

We are pleased to inform you that your manuscript has been judged scientifically suitable for publication and will be formally accepted for publication once it complies with all outstanding technical requirements.

With kind regards,

Baogui Xin, Ph.D.

Academic Editor

PLOS ONE
---

## [Editor Report · Acceptance letter]

22 May 2020

PONE-D-19-35078R2 

Participatory modelling for poverty alleviation using fuzzy cognitive maps and OWA learning aggregation 

Dear Dr. Papageorgiou:

I am pleased to inform you that your manuscript has been deemed suitable for publication in PLOS ONE. Congratulations! Your manuscript is now with our production department. 

With kind regards,

on behalf of

Professor Baogui Xin 

Academic Editor

PLOS ONE